# The CIN-TCP transcription factors regulate endocycle progression and pavement cell size by promoting cell wall pectin degradation

Feng Shen [1,2,3], He Zhang [1,2,3] ✉, Miaomiao Wan [1,2], Yanzhi Yang [1], Zheng Kuang[2], Liang Xiao [1,2], Daqing Zuo [1,2], Zhan Li[2], Genji Qin [2] & Lei Li [1] ✉

In plants, endoreplication, the process where nuclear DNA replicates in the absence of mitosis, and remodeling of the primary cell walls are both coupled with cell expansion. However, the mechanisms by which these two processes coordinate to determine cell size remain largely elusive. Here, employing the *tcpΔ7* septuple mutant disabling seven of the eight CIN-TCP transcription factors in Arabidopsis, we find that hindered endoreplication progression in *tcpΔ7* whereby ploidy increases from 8 C to beyond is correlated with an increase in cell wall pectin. CIN-TCPs transcriptionally activate *POLY-GALACTURONASE LIKE 1* (*PGL1*), which encodes a polygalacturonase down-regulating both abundance and molecular mass of pectin polymers. Genetic analysis of *PGL1* in both the wild type and *tcpΔ7* backgrounds confirm that pectin reduction promotes endocycle progression and cell enlargement. Collectively, these findings reveal a critical role of pectin in regulating endoreplication, providing insights in the understanding of cell growth and organ development in plants.

Leaves are the main photosynthesis organ, and their size determination is an important question in plant biology[1]. In flowering plants, as leaf cells exit the mitotic program and leave meristematic regions in leaf primordia, they differentiate and grow in volume. During this process, cells often undergo endoreplication in which DNA replication continues but chromosome separation and cell division are omitted, resulting in progressive increases in somatic ploidy levels[2,3]. As an alternative mode of cell cycle, the endocycle utilizes many elements of the mitotic cycle, including cyclins (CYC), cyclin-dependent kinases (CDKs), and CDK inhibitors, to regulate its onset, progression, and exit[4]. During the onset of the endocycle, whereby the ploidy level successfully increases to 8 C, the activity

of certain CYC-CDK complexes must be down-regulated[4,5]. For example, CELL CYCLE SWITCH 52 A (CCS52A) induces endocycle onset by activating the anaphase-promoting complex/cyclosome to degrade CYCs[6,7]. In contrast, the progression of the endocycle whereby the ploidy level increases from 8 C to 16 C and beyond is far less understood. Various studies have illustrated that endoreplication is associated with post-mitotic cell expansion and organ size determination. For example, ploidy levels were found to highly correlate with the size of pavement cells and trichomes[3,8]. In Arabidopsis mutants defective in topoisomerase VI, the endocycle is stalled at the 8 C stage and the mutant plants exhibit extreme dwarfism[9–12].

[1]Peking University Institute of Advanced Agricultural Sciences, Shandong Laboratory of Advanced Agricultural Sciences in Weifang, Shandong, China. [2]School of Advanced Agricultural Sciences and School of Life Sciences, Peking University, Beijing, China. [3]These authors contributed equally: Feng Shen, He Zhang. ✉e-mail: hez@pku.edu.cn; lei.li@pku-iaas.edu.cn

Different from metazoan cells, plant cells are enclosed in the primary cell walls that serve as a barrier, a structural support, and a site for cell-cell communications at the same time[13,14]. The cell wall is a highly dynamic structure composed of complex polysaccharides including mainly cellulose, hemicellulose, and pectin[13–15]. Pectin polymers, defined as polysaccharides rich in galacturonic acid, include homogalacturonan (HG), rhamnogalacturonan I, and the lesser abundant rhamnogalacturonan II[16–18]. Chemical and genetic analyses have shown that pectin is a major determinant of the chemical, structural, and mechanical properties of plant cell walls[16,19]. Molecular and nanoimaging studies have provided further support to the key regulatory roles of pectin in cell growth and tissue morphogenesis[20–22]. In addition to biosynthesis and modifications, degradation is also important for maintaining pectin homeostasis. Polygalacturonases are enzymes that hydrolyze the α-1,4 glycosidic bonds in galactosyluronic acid polymers and, therefore, can degrade HG[16,23]. Involvement of several polygalacturonases in regulating cell and tissue growth via reducing the molecular weight and/or abundance of HG has been reported[21,24].

Given their roles in cell size control, intrinsic associations between endoreplication and cell wall remodeling are anticipated. The observations that endocycle onset precedes rapid cell expansion and drastically impacts the expression of cell wall biosynthesis and modifying genes[25,26] have led to the proposal that endoreplication prepares cells for massive enlargement following cell cycle exit by promoting cell wall loosening[27]. Analysis of Arabidopsis plants with different somatic ploidy levels showed that incrementing ploidy was negatively correlated with pectin and hemicellulose contents in the cell wall[28]. A recent study showed that reduction of endoreplication enhances cell wall stiffening, actively reducing cell size[29]. However, whether cell wall remodeling has a direct role in regulating endocycle onset and/or progression remains elusive.

*TEOSINTE BRANCHED 1/CYCLOIDEA/PCF* (*TCP*) genes encode plant-specific transcription factors that play vital roles in controlling cell growth and plant architecture[30]. TCP proteins are classified as class I and class II based on the differences in the TCP domain, while class II is further divided into the CINCINNATA-like (CIN) clade and the CYCLOIDEA/TEOSINTE BRANCHED1-like (CYC/TB1) clade[30–32]. In Arabidopsis, the CIN-TCP clade consists of eight members (TCP2, 3, 4, 5, 10, 13, 17, and 24) that are crucial regulators of the mitotic cycle. In plants overexpressing microRNA319 (miR319), which targets five members of the CIN-TCP clade, prolonged maintenance of mitotic activity was observed, indicating that CIN-TCPs limit cell proliferation[33]. Moreover, reduction in the activities of these CIN-TCPs was shown to extend cell proliferation along leaf margins and result in large curled leaves made up of small cells, indicating that loss of CIN-TCP function increases cell number and decreases cell size[33,34]. Subsequently, *CYCLIN-DEPENDENT KINASE INHIBITOR 1* (*ICK1*), a repressor of the cell cycle, was identified as a direct downstream target of TCP4[35]. TCP24 was found to repress expression of components of the pre-replication complex required for the initiation of nuclear DNA replication and cell cycle progression from the G1 to the S phase[36]. Despite a study showing that induction of miR319-resistant TCP4 was able to commit leaf cells to exit proliferation and promote endoreplication[37], mechanistic investigations of the CIN-TCPs in coordinating endoreplication and cell wall remodeling remain scanty.

Here, we employed a *tcp2/3/4/5/10/13/17* septuple mutant, designated as *tcpΔ7*, which disabled seven of the eight members of the CIN-TCP clade in Arabidopsis[38], to study the relationship between endocycle and cell wall remodeling in cell size control. Our cell biology analyses showed that hindered endocycle progression in *tcpΔ7* was correlated with an increase in cell wall pectin. Molecular experiments revealed that the CIN-TCPs directly activate *POLYGALACTURONASE LIKE 1* (*PGL1*), which encodes a functional polygalacturonase that impacts both abundance and molecular mass of the pectin polymers. Genetic manipulations confirmed that pectin reduction specifically promotes endocycle progression but not onset to enlarge pavement cells. Collectively, these findings revealed a critical role of pectin in determining endocycle progression and cell growth, paving the way for further understanding organ development in plants.

## Results

### CIN-TCPs promote cell expansion and endoreplication

The CIN-TCP transcription factors exhibit high genetic redundancy in regulating plant morphogenesis[30,38]. To further explore their roles in controlling cell size, we employed the *tcp2/3/4/5/10/13/17* septuple mutant that disables seven of the eight members of the CIN-TCP clade in Arabidopsis[38] (Fig. 1a), which was referred to as *tcpΔ7* in this study. Cryo-scanning electron microscopy (cryo-SEM) analysis of cotyledons showed that the average pavement cell size in *tcpΔ7* was about 2.5 times smaller than that of the wild type, due to a reduction in the number of extremely large cells (Fig. 1b). This observation was consistent with the reported changes in cell size caused by miR319 overexpression that downregulated five members of the CIN-TCP clade[33]. Similar to the pavement cells, we found that the average size of cotyledon mesophyll cells in *tcpΔ7* was approximately 2.3 times smaller than that of the wild type (Fig. 1c). These observations indicate that CIN-TCPs positively regulate cell size.

Given that cell size is highly correlated with the DNA ploidy levels in pavement cells[3,8], we first compared the nuclear area between *tcpΔ7* and the wild type. DAPI staining of the nuclei of cotyledon pavement cells revealed that the nuclear area in *tcpΔ7* was significantly smaller, with a particular reduction in the number of large nuclei (Fig. 1d). Next, we performed flow cytometry analysis of the nuclei from the wild type and *tcpΔ7* cotyledons of the same age. Compared to the wild type, *tcpΔ7* exhibited a striking increase in the population of 2 C and 4 C nuclei and a concurrent reduction of 8 C, 16 C, and 32 C nuclei (Fig. 1e). Endoreplication index (EI) representing the average number of endocycles per nucleus has been used to quantitate endoreplication[30,39]. We found that the calculated EI of *tcpΔ7* was significantly decreased relative to the wild type (Fig. 1f). These results indicate that the endocycle is repressed in the *tcpΔ7* mutant.

To start the endocycle, cells need to escape from the mitotic cycle and transit from the 4 C stage to the 8 C stage, a process known as endocycle onset[4,5]. As the endocycle further advances, the ploidy level increases from 8 C to 16 C and beyond, a process known as endocycle progression[5]. To test whether both processes are impeded in *tcpΔ7*, we calculated the ratios of 8 C nuclei over 4 C nuclei (8 C/4 C) and 16 C nuclei over 8 C nuclei (16 C/8 C) as the surrogates for endocycle onset and progression, respectively[40,41]. We found that both the 8 C/4 C and 16 C/8 C ratios were significantly decreased in the *tcpΔ7* leaves in comparison to the wild type (Fig. 1g). These results indicate that the absence of CIN-TCPs hampers both endocycle onset and progression.

### CIN-TCPs regulate expression of genes related to both cell cycle and cell wall remodeling

To elucidate the potential mechanisms by which CIN-TCPs promote endocycle, we performed whole transcriptome RNA sequencing (RNA-seq) analysis of the wild type and *tcpΔ7* leaves. Compared to the wild type, 4685 genes were differentially expressed in *tcpΔ7*, including 2317 up-regulated and 2368 down-regulated (Supplementary Fig. 1a). We identified significantly enriched Gene Ontology (GO) terms associated with the differentially expressed genes to elucidate biological processes regulated by the CIN-TCPs. To deconvolute the highly redundant information among the GO terms, we performed clustering analysis and constructed an association network of the GO terms in the biological process category by calculating the similarity between each pair of GO terms[42] (Supplementary Fig. 1b).

We found that a cluster of GO terms related to the cell cycle was enriched in the constructed association network (Supplementary Figs. 1b and 2). Examination of individual genes showed that 19 of the

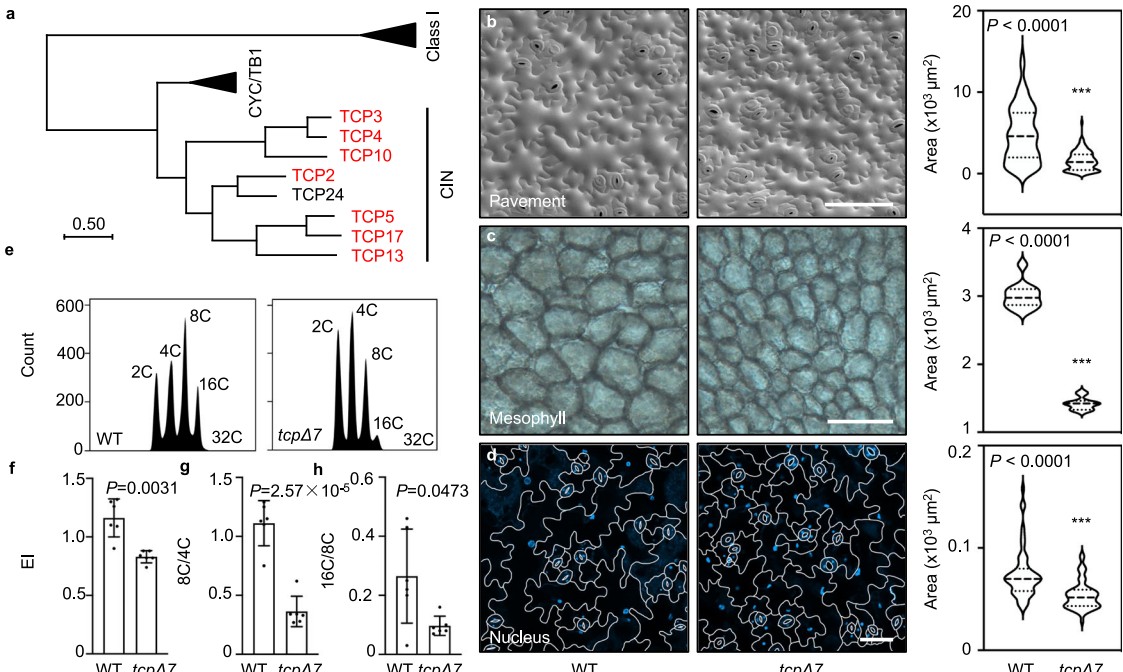

**Fig. 1 | CIN-TCPs promote cell expansion and endocycle. a** A phylogenic tree of the TCP genes in *Arabidopsis*. Disabled *CIN-TCP* genes in the *tcpΔ7* mutant are indicated in red. **b** Cryo-SEM analysis of the abaxial pavement cells in 7-day-old wild type and *tcpΔ7* cotyledons. Scale bar, 100 μm. Quantification of cell areas is shown on the right. Values are mean ± SD (*n* = 100 independent cells). Statistical analysis was performed using the Mann-Whitney test. Source data are provided as a Source Data file. **c** DIC microscopy analysis of the mesophyll cells in 7-day-old cotyledons. Scale bar, 100 μm. Quantification of cell areas is shown on the right. Values are mean ± SD (*n* = 300 independent cells). Statistical analysis was performed using a two-tailed unpaired Student's *t* test. Source data are provided as a Source Data file.

**d** DAPI staining of the nucleus in abaxial pavement cells of 10-day-old cotyledons. Scale bar, 50 μm. Quantification of nucleus areas is shown on the right. Values are mean ± SD (*n* = 55 independent cells). Statistical analysis was performed using the Mann-Whitney test. Source data are provided as a Source Data file. **e** Representative ploidy profiles obtained by flow cytometry analysis of cotyledons in 7-day-old wild type and *tcpΔ7* plants. Approximately 10,000 nuclei per sample were analyzed. C, haploid DNA content. **f**–**h** Quantification of EI (**f**), the ratio of 8 C to 4 C nuclei (**g**), and the ratio of 16 C to 8 C nuclei (**h**) from ploidy distributions. Values are mean ± SD (*n* = 6 biological replicates). Statistical analysis was performed using a two-tailed unpaired Student's *t* test. Source data are provided as a Source Data file.

49 *CYC* genes, 4 of the 14 *CDK* genes, and 2 of the 15 CDK inhibitors genes were differentially expressed in *tcpΔ7* (Supplementary Fig. 3a, b). This finding was corroborated by RT-qPCR analysis showing that selected genes encoding positive regulators of cell cycle were indeed up-regulated in *tcpΔ7*, whereas genes encoding negative regulators of cell cycle displayed the opposite trend (Supplementary Fig. 3c). In agreement with previously reported transcriptome profiling using the *rTCP4* plants that contains synonymous mutations in the miR319-binding site[35], transcriptome profiling using *tcpΔ7* revealed that CIN-TCPs broadly regulate genes related to the mitotic cell cycle. Thus, contrary to the previously reported mode of action for class I TCPs[43–45], CIN-TCPs may promote endocycle by inhibiting the mitotic cell cycle.

In addition to the cell cycle, a cluster of GO terms related to saccharide metabolism was conspicuously enriched in the constructed association network (Supplementary Figs. 1b and 4a). We postulated that this was in part due to enrichment of GO terms pertinent to the synthesis and metabolism of polysaccharides, the major cell wall component. Indeed, GO terms directly related to cell wall biogenesis and organization, including cell wall polysaccharide metabolism and cell wall pectin metabolism, were significantly associated with genes regulated by CIN-TCPs (Supplementary Fig. 4b). These results suggest that CIN-TCPs regulate the expression of genes related to cell wall remodeling.

### CIN-TCPs modulate cell wall pectin level and PG activity
The observed transcriptomic changes prompted us to analyze cell wall polysaccharides. Chemical quantification of cellulose and hemicellulose contents in the rosette leaves showed no significant differences between *tcpΔ7* and the wild type (Fig. 2a). In contrast, total pectin level in *tcpΔ7* leaves significantly increased in comparison to the

wild type (Fig. 2a), indicating that CIN-TCPs specifically repress the accumulation of cell wall pectin. To corroborate this conclusion, we performed immunolabeling assays to examine the accumulation of HG that accounts for the majority of pectin in plant cell walls[16]. We used the 2F4 and LM19 monoclonal antibodies, which detects $Ca^{2+}$-cross-linked HG and de-methylesterified HG, respectively[46,47]. Fluorescence microscopy analysis revealed that the intensity of the LM19 (Fig. 2b, c) and 2F4 signals (Fig. 2d, e) was both significantly enhanced in *tcpΔ7* cotyledons and young rosette leaves compared to the wild type. By contrast, the signals of Fluorescent Brightener 28 (FB28), which mainly stains cellulose, showed no substantial difference between *tcpΔ7* and the wild type (Fig. 2b–e). These observations confirmed that CIN-TCPs repress cell wall pectin abundance.

Variation in pectin content is attributed to changed pectin biosynthesis and/or degradation[23,48]. As de-methylesterified pectin could be crosslinked by calcium and/or degraded by PGs[21], increase of both de-methylesterified HG and $Ca^{2+}$-crosslinked HG in *tcpΔ7* prompted us to examine whether in vivo PG activity was altered in *tcpΔ7*. Using an established assay for PG activity using a polygalacturonic acid substrate as previously reported[21,49], we found that total protein extracted from *tcpΔ7* rosette leaves possessed a significantly lower PG activity than the wild type (Fig. 2f). This result further suggests that CIN-TCPs repress cell wall pectin accumulation possibly via upregulating total PG activity in the leaves.

### CIN-TCPs directly bind to the *PGL1* promoter and activate its expression
To pinpoint which genes contribute to the reduction of PG activity in *tcpΔ7*, we profiled the expression of all 68 *PG*-related genes annotated

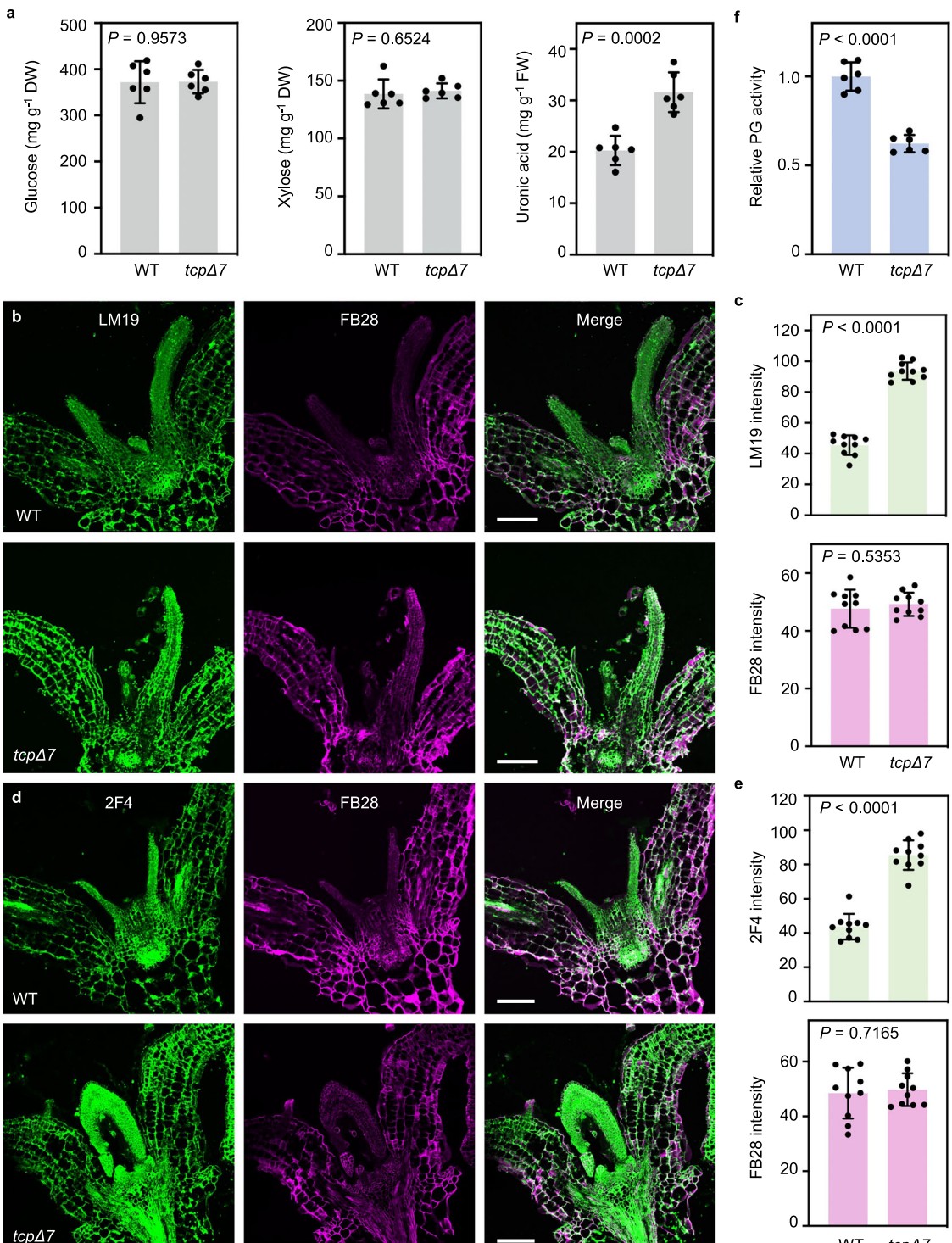

**Fig. 2 | *tcpΔ7* exhibits elevated cell wall pectin and reduced PG activity.**
**a** Chemical quantification of the saccharide derivatives representing the three cell wall polysaccharides. Wall polysaccharides were extracted from the third pair of rosette leaves of 4-week-old wild-type and *tcpΔ7* plants. Values are mean ± SD (*n* = 6 independent experiments). Statistical analysis was performed using a two-tailed unpaired Student's *t* test. Source data are provided as a Source Data file. DW, dry weight. FW, fresh weight. Source data are provided as a Source Data file.
**b**−**e** Immunolabeling analysis of wild type and *tcpΔ7* cell wall pectin. Sections of 7-day-old seedlings were stained with LM19 to label de-methylesterified HG (**b**), 2F4

to label Ca²⁺-crosslinked de-methylesterified HG (**d**), and FB28 to detect cell wall cellulose. Scale bars, 100 μm. Average intensities of LM19 and 2F4, along with that of FB28, are shown in (**c** and **e**), respectively. Values are mean ± SD of 10 areas (100 μm by 100 μm) from five cotyledons. Statistical analysis was performed using a two-tailed unpaired Student's *t* test. Source data are provided as a Source Data file. **f** Comparison of in vivo PG activity of 7-day-old wild type and *tcpΔ7* seedlings. Values are mean ± SD (*n* = 6 independent experiments) of measured absorbance normalized to the wild type. Statistical analysis was performed using a two-tailed unpaired Student's *t* test. Source data are provided as a Source Data file.

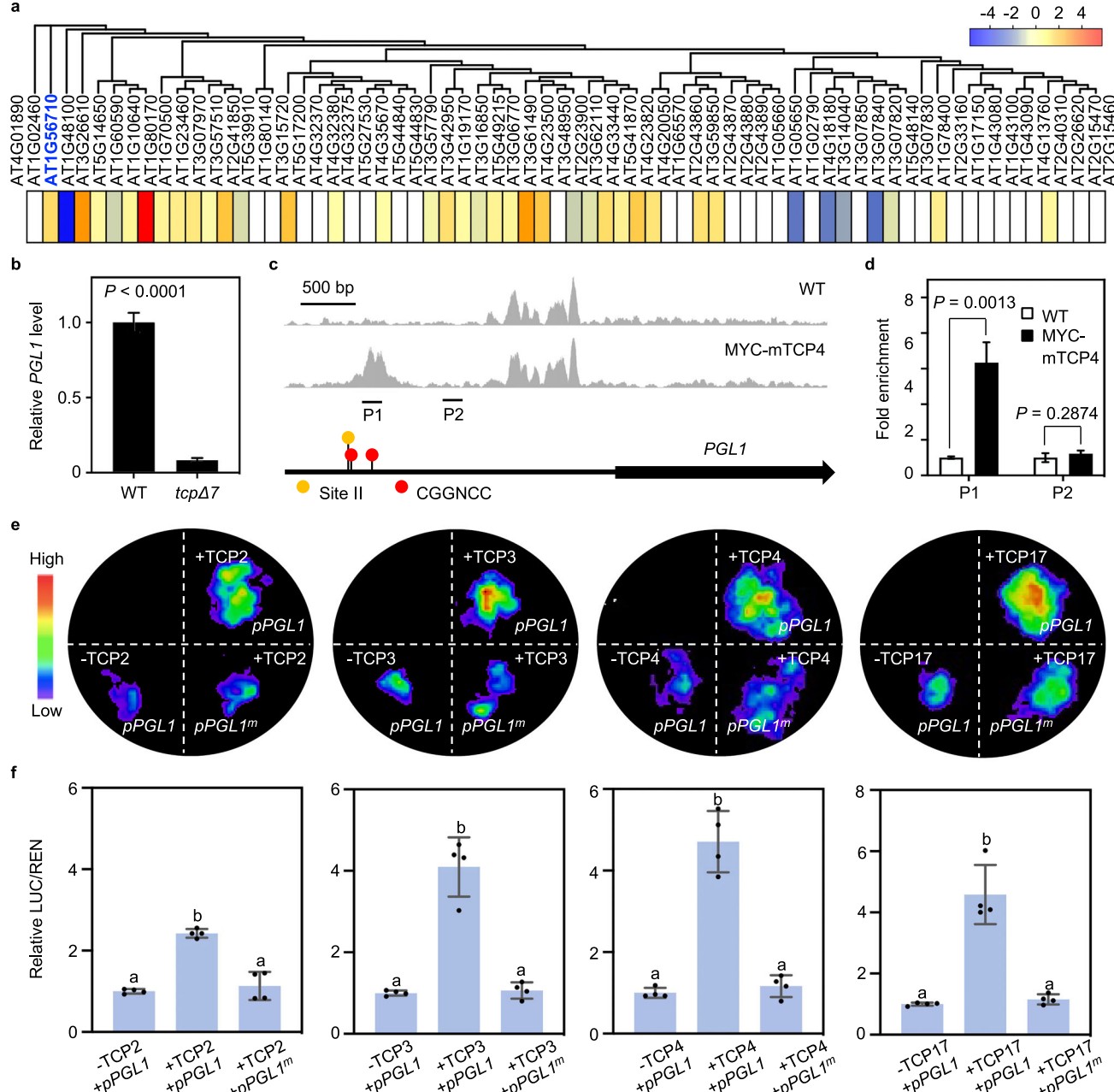

**Fig. 3 | TCP4 directly activates *PGL1* expression. a** Heatmap showing relative transcript levels of the 68 *PG* genes in *tcpΔ7* determined by RNA-seq. Colors represent log₂ fold change in *tcpΔ7* compared to the wild type. **b** Relative transcript abundance of *PGL1* in the wild type and *tcpΔ7* cotyledons determined by RT-qPCR analysis using *ACTIN7* as the internal control. Values are mean ± SD (*n* = 3 biological replicates). Statistical analysis was performed using a two-tailed unpaired Student's *t* test. Source data are provided as a Source Data file. **c** TCP4 binding profiles at the *PGL1* locus based on ChIP-seq analysis of the wild type and 35S-MYC-mTCP4 plants. The black arrow represents the genomic region of *PGL1* and the transcriptional direction. The yellow and red dots indicate the site II and CGGNCC cis-regulatory motifs recognized by TCP4, respectively. **d** ChIP-qPCR analysis of 7-day-old wild type and *35S-MYC-mTCP4* seedlings. Immunoprecipitated DNA using a MYC antibody was subjected to qPCR analysis. Positions of the P1 and P2 amplicons are

indicated in (**c**). Values are mean ± SD (*n* = 3 biological replicates) of enrichment relative to the wild type. \*\*, Statistical analysis was performed using a two-tailed unpaired Student's *t* test. Source data are provided as a Source Data file. **e** Dual luciferase assay to test the effects of CIN-TCPs on the *PGL1* promoter. The *pPGL1:LUC-35S:REN* or *pPGL1ᵐ:LUC-35S:REN* reporter was co-infiltrated with the *35S:TCP2/3/4/17* effectors (+) or the empty vector alone (-) into tobacco leaf epidermal cells and imaged for LUC activity. **f** Quantification of the LUC/REN ratio from the *pPGL1:LUC-35S:REN* or *pPGL1ᵐ:LUC-35S:REN* reporter in the presence and absence of the effectors. Values are mean ± SD (*n* = 4 independent experiments) normalized to the control. Statistical analysis was performed using one-way ANOVA with Tukey's multiple comparison test, and different letters above the bars indicate statistical significance at *p* < 0.001. Source data are provided as a Source Data file.

in Arabidopsis[50]. Based on the RNA-seq data, we found that 12 of the *PG* genes were differentially expressed in *tcpΔ7* relative to the wild type (Fig. 3a). Among these, *POLYGALACTURONASE LIKE 1* (*PGL1*; AT1G56710) showed the most drastic decrease in transcript abundance in *tcpΔ7* (Fig. 3a). Using RT-qPCR, we confirmed that the *PGL1*

transcript level was repressed by more than 10-fold in the *tcpΔ7* cotyledons compared to the wild type (Fig. 3b).

To determine whether *PGL1* is a direct target gene of CIN-TCPs, we analyzed the previously reported chromatin immunoprecipitation sequencing (ChIP-seq) data for TCP4, one of the CIN-TCPs. The ChIP-

seq was performed using *35S:MYC-mTCP4* seedlings in which the Cauliflower Mosaic Virus *35S* promoter was used to drive *mTCP4*, a mutated form of TCP4 that only alters the coding region to make it resistant to miR319-mediated downregulation[51]. We found a TCP4-specific binding peak in the *PGL1* promoter, approximately 2 Kb upstream of the transcription start site (Fig. 3c). Scanning this region revealed three putative *cis*-regulatory elements that have been reported as binding sites for CIN-TCPs[30,52], including one site II element and two CGGNCC elements (Fig. 3c). Using ChIP-qPCR assay performed on the *35S:MYC-mTCP4* seedlings, significant enrichment of TCP4 occupancy at the *PGL1* promoter encompassing the proximal CGGNCC element was confirmed (Fig. 3d), indicating that TCP4 binds to the *PGL1* promoter.

To assess the effect of CIN-TCPs on the *PGL1* promoter, we employed the firefly luciferase (LUC) and *Renilla* luciferase (REN) dual reporter system[53] in a transient expression assay. We generated the *35S:TCP2*, *35S:TCP3*, *35S:TCP4*, and *35S:TCP17* effector constructs. We then used the *PGL1* promoter to drive *LUC* expression and generated the *pPGL1:LUC-35S:REN* dual reporter construct (Supplementary Fig. 5). We tested the four effector-reporter combinations through co-infiltration of tobacco (*Nicotiana benthamiana*) leaf epidermal cells. We found that, in the presence of TCP2/3/4/17, the LUC chemiluminescence was all markedly increased (Fig. 3e) and the ratio of LUC/REN intensities significantly elevated (Fig. 3f), indicating that the tested CIN-TCPs positively regulate the *PGL1* promoter. Moreover, we mutated the three *cis*-regulatory elements (site II, CGGNCC-I and CGGNCC-II) to generate the *pPGL1^m:LUC-35S:REN* reporter construct (Supplementary Fig. 5). We found that both the LUC intensities (Fig. 3e) and the LUC/REN ratios (Fig. 3f) in the presence of TCP2/3/4/17 were markedly decreased comparing to *pPGL1:LUC-35S:REN*.

Biolayer interferometry (BLI) is a label-free optical biosensing technology that analyzes biomolecular interactions in real-time[54]. Employing BLI, we monitored the binding kinetics between recombinant TCP2 or TCP3 and DNA fragments containing the three *cis*-regulatory elements in the *PGL1* promoter (Supplementary Fig. 6). We found that all the combinations between TCP2 or TCP3 and the DNA elements showed high binding affinities with the dissociation constants consistently in the ~ 0.1 μM range. In contrast, mutating these elements abolished the binding by TCP2 or TCP3 (Supplementary Fig. 6). Taken together, our results demonstrated that CIN-TCPs are able to directly bind to the promoter of *PGL1* and activate its expression.

## PGL1 is a functional PG positively regulating cell size and endocycle progression

To investigate the biological role of *PGL1*, we identified an *Arabidopsis* T-DNA line carrying an insertion in the first exon of *PGL1*, which was named *pgl1-1*, and generated a deletion allele based on CRISPR/Cas9, which was named *pgl1-2* (Supplementary Fig. 7a). We also generated *PGL1*-overexpressing plants (*PGL1-OX*) in which the enhanced *35S* promoter was used to drive expression of the *PGL1* coding region. RT-qPCR analysis showed that *PGL1* expression was drastically compromised in *pgl1* and enhanced in *PGL1-OX* seedlings compared to the wild type, respectively (Fig. 4a, b and Supplementary Fig. 7b, c). The *pgl1* mutants displayed significantly reduced cotyledon areas in comparison to the wild type, whereas *PGL1-OX* displayed the opposite phenotype (Fig. 4a, c and Supplementary Fig. 7d, e). Moreover, cryo-SEM analysis showed that the *pgl1* cotyledons exhibited significantly smaller average area of pavement cells compared to the wild type, with a concurrent reduction of the number of extremely large cells while *PGL1-OX* plants exhibited the opposite phenotypes (Fig. 4d, e), indicating that *PGL1* is a positive regulator of pavement cell size. To determine whether the effect of *PGL1* on cell size was related to PG activity, we examined the in vivo PG activity in *PGL1-OX* and *pgl1*. We found that the total protein extracted from *PGL1-OX* seedlings

possessed a significantly higher PG activity than the wild type, whereas the *pgl1* seedlings displayed the opposite phenotype (Fig. 4f). These results indicate that PGL1 is likely a functional PG that substantially contributes to the total in vivo PG activity in the leaves. We further quantified the total cell wall pectin contents in *PGL1-OX* and *pgl1* seedlings, and a significant decrease in *PGL1-OX* and a significant increase in *pgl1* relative to the wild type were observed (Fig. 4g). Thus, *PGL1* positively regulates cell expansion by increasing total PG activity to limit pectin abundance in the cell walls.

To determine whether *PGL1* impacts endoreplication, we performed flow cytometry analysis of the *pgl1* and *PGL1-OX* nuclei. We sampled the third pair of rosette leaves at three different development stages: stage 1 at 14 days after sowing (DAS) in which the cells are primarily in the mitotic cycles, stage 2 at 18 DAS in which the cells exit the mitotic cycle, and stage 3 at 28 DAS in which the cells are endoreduplicating[55,56]. Compared to the wild type at the same ages, *pgl1* showed a striking decrease in the population of 16 C nuclei only at stage 3 when cells were fully committed to endoreduplication (Fig. 5a–c), which was in accordance with the observation that *pgl1* had a reduced number of large pavement cells (Fig. 4d). On the contrary, *PGL1-OX* showed an increase in the population of 16 C nuclei over the wild type only at stage 3 (Fig. 5c), also in accordance with its increased number of large pavement cells (Fig. 4d). Examination of leaf morphology at the three stages confirmed that leaf area and cell number of *pgl1* and *PGL1-OX* remained the same as the wild type until stage 3 (Fig. 5d, e). Together with the calculated EI that was significantly lower in *pgl1* but higher in *PGL1-OX* only at stage 3 (Fig. 5a–c), these results indicate that *PGL1* promotes endoreplication. To further discern whether *PGL1* impacts endocycle onset or progression, we calculated the 8 C/4 C and 16 C/8 C ratios at stage 3 (Fig. 5c). We found that the 8 C/4 C ratios showed no significant difference between *pgl1* or *PGL1-OX* against the wild type (Fig. 5c). On the contrary, the 16 C/8 C ratio was significantly decreased in the *pgl1* but significantly increased in *PGL1-OX* leaves relative to the wild type (Fig. 5c). These results demonstrated that *PGL1* positively regulates endoreplication by specifically promoting endocycle progression.

## PGL1 rescues the endocycle progression defect in tcpΔ7

To test whether *PGL1* is sufficient to rescue the *tcpΔ7* phenotypes, we generated the *tcpΔ7 PGL1-OX* plant. RT-qPCR analysis showed that the *PGL1* transcript level was drastically enhanced in independent *tcpΔ7 PGL1-OX* lines compared to *tcpΔ7* (Supplementary Fig. 8a). While *PGL1* did not affect the jagged and curly shape of the *tcpΔ7* cotyledons[38], the already larger than wild type cotyledon size of *tcpΔ7* was further enlarged in *tcpΔ7 PGL1-OX* (Fig. 6a and Supplementary Fig. 8b, c). Moreover, cryo-SEM analysis showed that the area of pavement cells in *tcpΔ7 PGL1-OX* was significantly larger than *tcpΔ7*, albeit not to the same size as the wild type (Fig. 6b, c). These results indicate that *PGL1*-mediated cell enlargement is independent of the enhanced cell proliferation defect that causes overgrowth of the leaf margins in *tcpΔ7*[33,34,38,57].

To determine how endocycle is impacted in *tcpΔ7 PGL1-OX*, we compared ploidy levels among the wild type, *tcpΔ7,* and *tcpΔ7 PGL1-OX* plants. Flow cytometry analysis revealed that the proportion of nuclei with high ploidy levels was increased in *tcpΔ7 PGL1-OX,* with a calculated EI significantly higher than *tcpΔ7* (Fig. 6d, e). Interestingly, whilst the 8 C/4 C ratio in *tcpΔ7 PGL1-OX* showed no obvious change compared to *tcpΔ7*, the 16 C/8 C ratio was significantly higher than *tcpΔ7* and restored to the same level as the wild type (Fig. 6f). These results indicate that overexpression of *PGL1* in the *tcpΔ7* mutant specifically rescued the cell enlargement and endocycle progression defects.

We further tested whether *PGL1* rescues endocycle progression in *tcpΔ7* by acting on pectin. Quantification of extracted cell wall polysaccharides revealed that elevated pectin content in the *tcpΔ7* seedlings was significantly lowered in *tcpΔ7 PGL1-OX* (Fig. 7a).

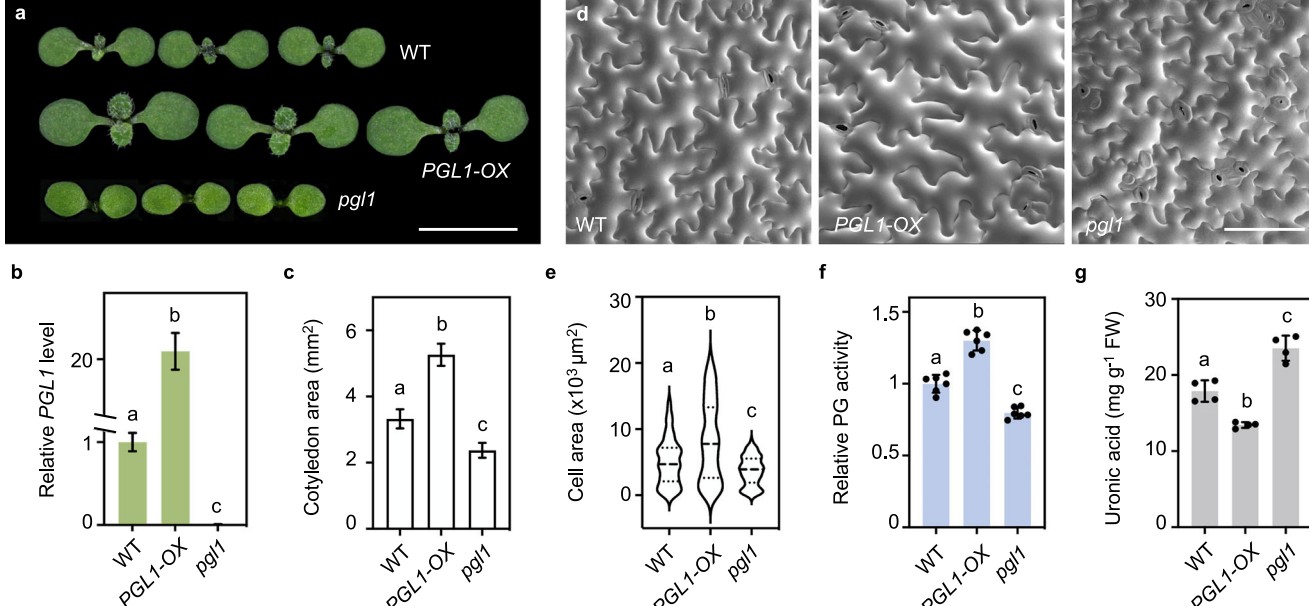

**Fig. 4 | *PGL1* promotes cell expansion and represses cell wall pectin.**
**a** Morphology of 7-day-old seedlings of the indicated genotypes. Scale bar, 0.5 cm.
**b** Relative transcript abundance of *PGL1* in the cotyledons determined by RT-qPCR analysis using *ACTIN7* as the internal control. Values are mean ± SD (*n* = 3 biological replicates). Statistical analysis was performed using one-way ANOVA with Tukey's multiple comparison test, and different letters above the bars indicate statistical significance at *p* < 0.001. Source data are provided as a Source Data file.
**c** Quantification of cotyledon areas. Values are mean ± SD (*n* = 30 independent cotyledons). Statistical analysis was performed using one-way ANOVA with Tukey's multiple comparison test, and different letters above the bars indicate statistical significance at *p* < 0.001. Source data are provided as a Source Data file. **d** Cryo-SEM analysis of the adaxial pavement cells in 7-day-old cotyledons. Scale bar, 100 μm.
**e** Quantification of pavement cell areas. Values are mean ± SD (*n* = 100 independent

cells). Statistical analysis was performed using the Kruskal-Wallis test, and different letters indicate genotypes with significant differences at *p* < 0.01. Source data are provided as a Source Data file. **f** Comparison of in vivo PG activity in 7-day-old seedlings of the indicated genotypes. Values are mean ± SD (*n* = 6 independent experiments) of measured absorbance normalized to the wild type. Statistical analysis was performed using one-way ANOVA with Tukey's multiple comparison test, and different letters above the bars indicate statistical significance at *p* < 0.001. Source data are provided as a Source Data file. **g** Quantification of uronic acid contents in cell walls of 7-day-old seedlings. Values are mean ± SD (*n* = 4 independent experiments). Statistical analysis was performed using one-way ANOVA with Tukey's multiple comparison test, and different letters above the bars indicate statistical significance at *p* < 0.01. FW, fresh weight. Source data are provided as a Source Data file.

Immunolabeling analysis using the 2F4 antibody showed that the *tcpΔ7 PGL1-OX* cotyledons had significantly weaker labeling than that in *tcpΔ7* (Fig. 7b, c). We performed fast protein liquid chromatography (FPLC) coupled with uronic acid detection to compare the molecular mass of pectin polymers in the wild type, *tcpΔ7,* and *tcpΔ7 PGL1-OX* seedlings. Consistent with previous reports in Arabidopsis[24,49,58], the pectin polymer peak eluted from the wild type cell wall exhibited a molecular mass of approximately 150 kD (Fig. 7d). In contrast, the eluted pectin polymer peak in *tcpΔ7* exhibited a molecular mass well over 200 kD (Fig. 7d). In *tcpΔ7 PGL1-OX*, the pectin molecular mass was near 200 kD, in between that of the *tcpΔ7* and the wild type (Fig. 7d). In addition, the amount of pectin estimated by the area of the eluted peaks was in agreement with the chemical quantification results (Fig. 7a). The relative PG activity in *tcpΔ7 PGL1-OX* was significantly higher than that in *tcpΔ7* (Fig. 7e). Taken together, these results indicate that *PGL1* overexpression in the *tcpΔ7* background led to substantial reduction of the abundance and molecular mass of the pectin polymers by increasing the PG activity.

**Pectin regulating miR775 promotes endocycle progression**
To independently verify the relationship among pavement cell size, wall pectin, and endocycle progression, we employed a pectin-regulating microRNA, miR775, which post-transcriptionally silences a galactosyltransferase gene involved in maintaining pectin content and cell wall elastic modulus[22]. We have previously shown that overexpression of miR775 in Arabidopsis (*MIR775-OX*) resulted in reduced cell wall pectin level and enlarged epidermal cells relative to the wild type[22]. Using cryo-SEM, we confirmed that *MIR775-OX* exhibited

significantly larger cotyledon pavement cells in comparison to the wild type (Fig. 8a). In addition, using chemical quantification and LM19 immunolabelling, we confirmed that pectin content in *MIR775-OX* was significantly lower than the wild type (Supplementary Fig. 9).

DAPI staining of cotyledon pavement cells showed that the nuclear size in *MIR775-OX* was significantly larger than that of the wild type (Fig. 8b). Flow cytometry analysis further showed that the number of cells with the 16 C and 32 C ploidy levels was significantly increased in *MIR775-OX* leaves (Fig. 8c). The calculated EI in *MIR775-OX* was significantly higher than that of the wild type (Fig. 8d). Moreover, we found that the 8 C/4 C ratios exhibited no difference between *MIR775-OX* and the wild type, whereas the 16 C/8 C ratio was increased significantly in *MIR775-OX* (Fig. 8d). Taken together, these results indicate that miR775 is able to repress wall pectin accumulation, promote endocycle progression, and enlarge pavement cells.

**Hindering endocycle onset does not alter cell wall pectin level**
To further confirm the association of pectin level with endocycle progression but not its onset, we employed the *cell cycle switch protein S2 a2 (ccs52a2)* mutant that disrupts CCS52A2, a rate-limiting activator of the anaphase promoting complex/cyclosome E3 ubiquitin ligase that promotes exit from the mitotic cell cycle and entry into the endocycle[7,59]. Flow cytometry analysis confirmed that *ccs52a2* cotyledon cells contain a higher proportion of 2 C and 4 C nuclei but a lower proportion of 8 C and 16 C nuclei than the wild type (Fig. 9a). The calculated EI of *ccs52a2* was significantly lower than the wild type (Fig. 9b). Moreover, the 8 C/4 C ratio in *ccs52a2* exhibited a significant decrease while the 16 C/8 C ratio showed no difference in comparison

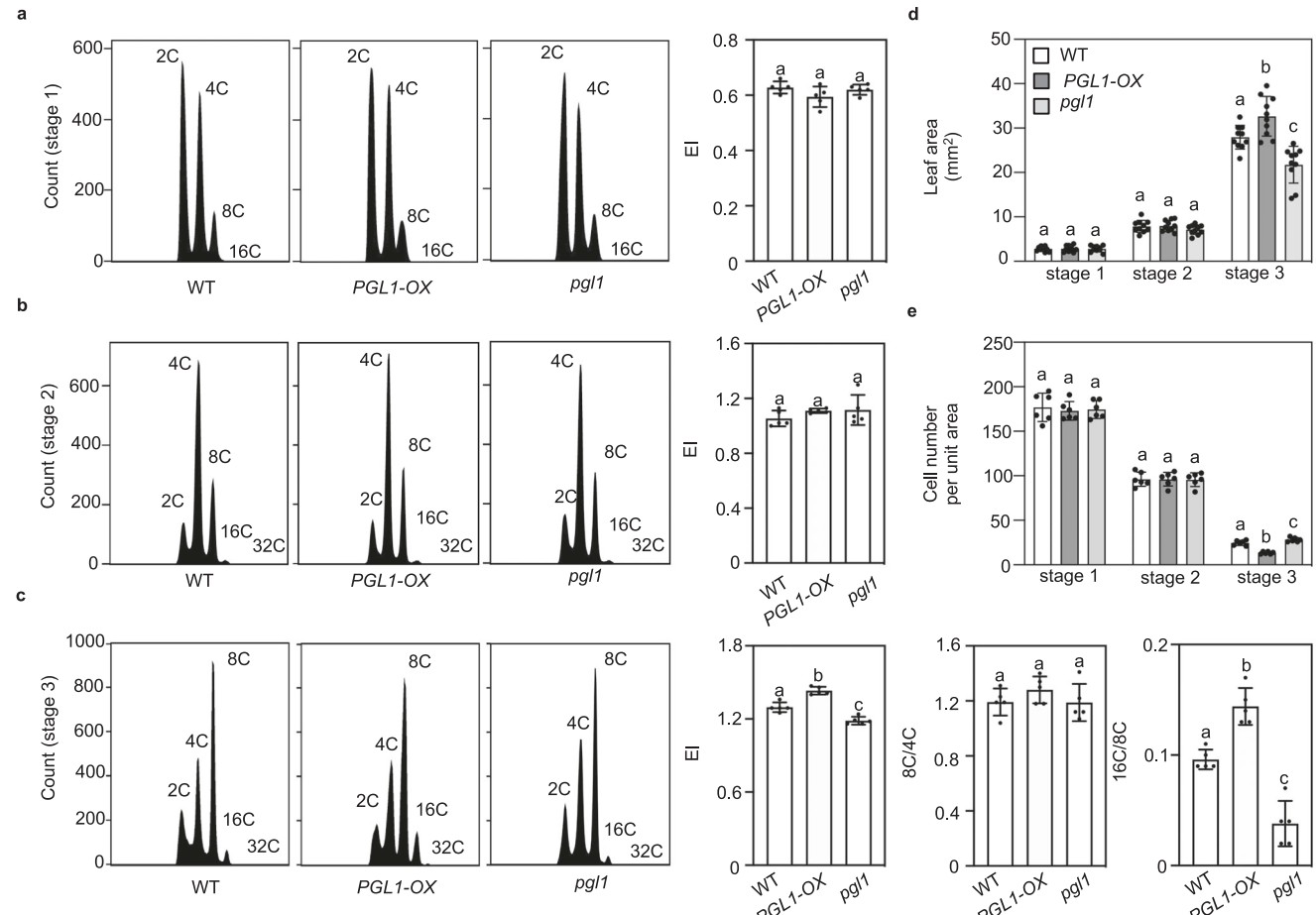

**Fig. 5 | *PGL1* promotes endocycle progression. a–c** Representative ploidy profiles of the third pair of leaves of the indicated genotypes at three leaf development stages. Stage 1 (**a**), stage 2 (**b**), and stage 3 (**c**) were sampled at 14, 18, and 28 DAS, respectively. Quantification of parameters related to endocycle in the respective stages is shown on the right. Values are mean ± SD (*n* = 5 biological replicates). Statistical analysis was performed using one-way ANOVA with Tukey's multiple comparison test, and different letters above the bars indicate statistical significance at *p* < 0.01. Source data are provided as a Source Data file. **d** Quantification of leaf area of the indicated genotypes at three leaf stages. Values are mean ± SD (*n* = 10

biological replicates). Statistical analysis was performed using one-way ANOVA with Tukey's multiple comparison test, and different letters above the bars indicate statistical significance at *p* < 0.05. Source data are provided as a Source Data file. **e** Quantification of the number of epidermal cells per unit area (366 by 366 μm²) at three leaf stages in the indicated genotypes. Values are mean ± SD (*n* = 6 biological replicates). Statistical analysis was performed using one-way ANOVA with Tukey's multiple comparison test, and different letters above the bars indicate statistical significance at *p* < 0.05. Source data are provided as a Source Data file.

to the wild type (Fig. 9c). Thus, consistent with previous reports[59,60], endocycle onset but not progression is effectively hindered in *ccsS2a2*. However, quantification of pectin levels showed no significant difference between *ccsS2a2* and wild-type cotyledons (Fig. 9d). In addition, *ccsS2a2* possessed comparable PG activity as the wild type (Fig. 9e). Together, these results indicate that hindering endocycle onset does not alter pectin abundance in the cell walls.

## Discussion

Plant cells are surrounded by a primary cell wall that needs to be loosened in a controlled manner for nondestructive cell expansion during various biological processes[19,61]. In this study, we found that two independent molecular routes regulating cell wall pectin both impact endocycle progression and cell enlargement (Fig. 10). In the first route, we found that the CIN-TCP transcription factors, in addition to regulating endocycle onset (Supplementary Figs. 1–3), promote endoreplication to advance beyond the 8 C stage by specifically modulating cell wall pectin metabolism (Figs. 1, 2). We showed that CIN-TCPs downregulate pectin abundance and molecular sizes by directly binding to the promoter and activating transcription of *PGL1* (Fig. 3). Genetic analysis confirmed PGL1 as a

functional PG and its overexpression results in significantly reduced pectin content, enlarged pavement cells, and restoration of endocycle progression in *tcpΔ7* (Figs. 4–7). In the second route, we found that miR775, which post-transcriptionally silences a galactosyltransferase gene required for maintaining pectin content, the elastic modulus of the cell wall, and epidermal cell sizes[22], promotes progression of endoreplication (Fig. 8). These results collectively support a working model in which modulation of cell pectin level is sufficient and necessary to alter endocycle progression and cell enlargement (Fig. 10).

Although further studies are needed, our findings provide fresh insights into the relationship among pectin, endocycle progression, and cell enlargement. Foremost, we found that cell wall pectin drives endocycle progression, and this regulation is independent of endocycle onset based on several lines evidence. For instance, we found that in the *tcpΔ7*, *pgl1*, and *MIR775-OX* plants with altered pectin levels, endocycle progression but not onset was affected (Figs. 4–8). On the contrary, in the *ccsS2a2* mutant with defective endocycle onset, cell wall pectin was not altered in comparison to the wild type (Fig. 9). Moreover, when *PGL1* was overexpressed in *tcpΔ7*, the total PG activity, cell wall pectin level, endocycle progression, and growth of pavement

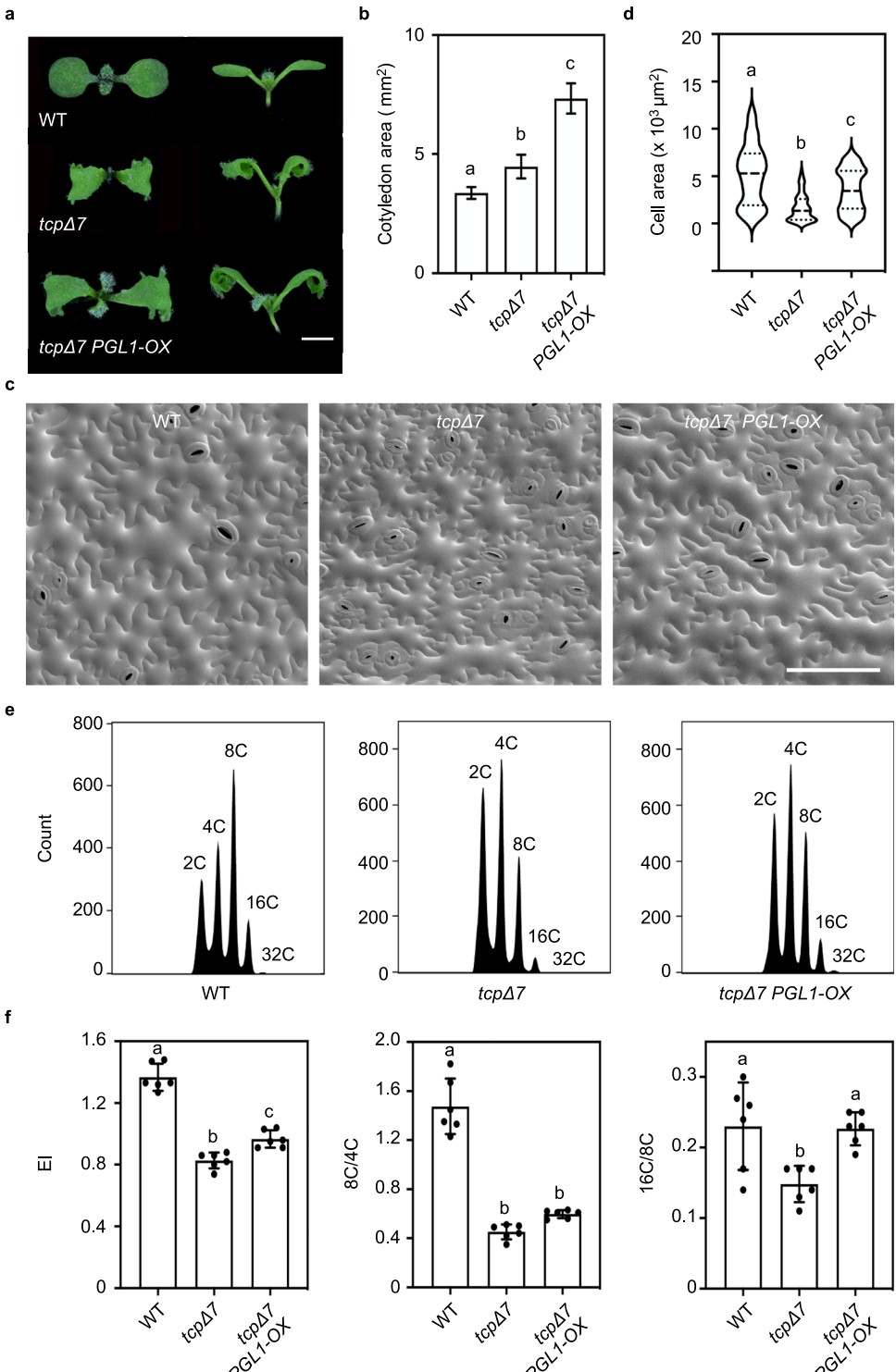

**Fig. 6 | *PGL1* rescues the endocycle progression defect in *tcpΔ7*. a** Morphology of 7-day-old seedlings of the indicated genotypes. Scale bar, 0.2 cm. **b** Quantification of cotyledon sizes. Values are mean ± SD ($n = 30$ independent cotyledons). Statistical analysis was performed using one-way ANOVA with Tukey's multiple comparison test, and different letters above the bars indicate statistical significance at $p < 0.01$. Source data are provided as a Source Data file. **c** Cryo-SEM analysis of the abaxial pavement cells in 7-day-old cotyledons. Scale bar, 100 μm. **d** Quantification of pavement cell areas. Values are mean ± SD ($n = 100$ independent cells). Statistical analysis was performed using the Kruskal-Wallis test, and different letters above the bars indicate statistical significance at $p < 0.01$. Source data are provided as a Source Data file. **e** Representative ploidy profiles of the third pair of leaves in 4-week-old plants. **f** Quantification of parameters related to endocycle using ploidy profiles obtained by flow cytometry analysis. Values are mean ± SD ($n = 6$ independent experiments). Statistical analysis was performed using one-way ANOVA with Tukey's multiple comparison test, and different letters above the bars indicate statistical significance at $p < 0.001$. Source data are provided as a Source Data file.

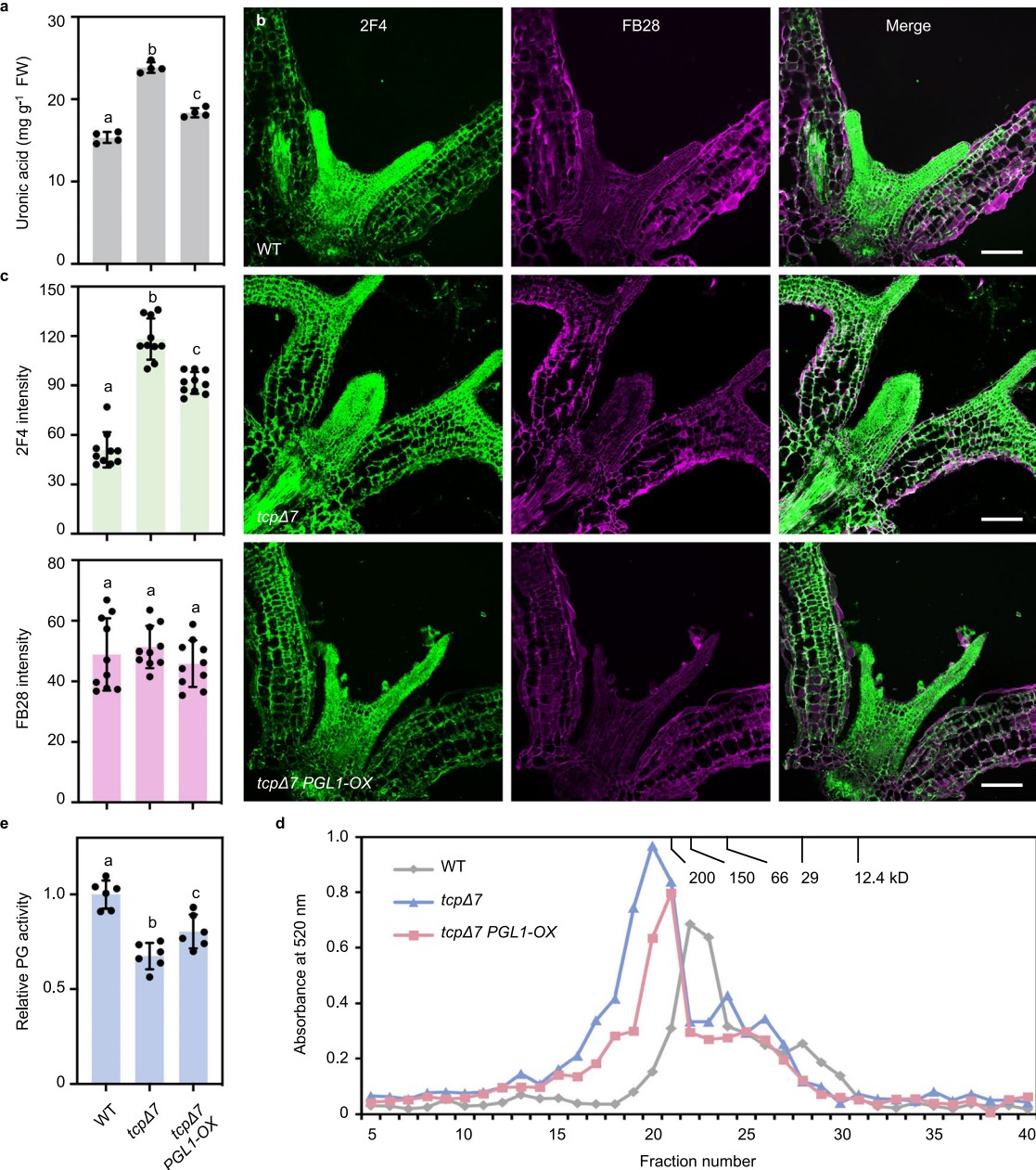

**Fig. 7 | *PGL1* partially restores cell wall pectin accumulation in *tcpΔ7*.**
**a** Quantification of cell wall pectin in 7-day-old seedlings of the indicated geno-
types. Values are mean ± SD (*n* = 4 independent experiments). Statistical analysis
was performed using one-way ANOVA with Tukey's multiple comparison test, and
different letters above the bars indicate statistical significance at *p* < 0.001. FW,
fresh weight. Source data are provided as a Source Data file. **b** Immunolabeling
analysis of pectin by 2F4 and FB28 staining. Scale bars, 100 μm. **c** Quantification of
average intensity of 2F4 and FB28. Values are mean ± SD of 10 areas (100 μm by
100 μm) from five cotyledons. Statistical analysis was performed using one-way
ANOVA with Tukey's multiple comparison test, and different letters above the bars
indicate statistical significance at *p* < 0.001. Source data are provided as a Source

Data file. **d** Molecular mass analysis of CDTA-soluble pectin extracted from 7-day-
old seedlings of the indicated genotypes by size-exclusion chromatography. Uronic
acid content in fractions 5 to 40 was measured by absorbance at 520 nm. Molecular
masses of the standards used to calibrate the column are indicated on top. Source
data are provided as a Source Data file. **e** Comparison of in vivo PG activity in 7-
day-old seedlings of the indicated genotypes. Values are mean ± SD (*n* = 6 independent
experiments) of measured absorbance normalized to the wild type. Statistical
analysis was performed using one-way ANOVA with Tukey's multiple comparison
test, and different letters above the bars indicate statistical significance at *p* < 0.05.
Source data are provided as a Source Data file.

cells were effectively restored (Figs. 6, 7). However, the jagged, curly,
and enlarged leaf morphology due to increased cell proliferation in
*tcpΔ7* was not affected in *tcpΔ7 PGL1-OX* (Fig. 6)[33,34,38,57]. In fact, with cell
enlargement restored, the *tcpΔ7 PGL1-OX* plants exhibited even larger
leaves than *tcpΔ7* (Fig. 6 and Supplementary Fig. 8). Thus, pectin
metabolism specifically regulates endocycle progression with an effect
on cell enlargement and leaf growth additive to that bestowed by cell
proliferation.

Our findings imply that ploidy level and cell wall pectin are
coordinated in 8 C cells to determine whether the endocycle will fur-
ther advance. Given that an increase in ploidy greatly contributes to
cell enlargement, especially the epidermal cells[3,8], the accompaniment
of endocycle progression with relaxation of the cell wall and changes in
mechanical properties of the wall are expected[19,61]. It could be extra-
polated from our model that cell wall properties bestowed by different
pectin levels would be different in 4 C, 8 C, and 16 C cells. Although a

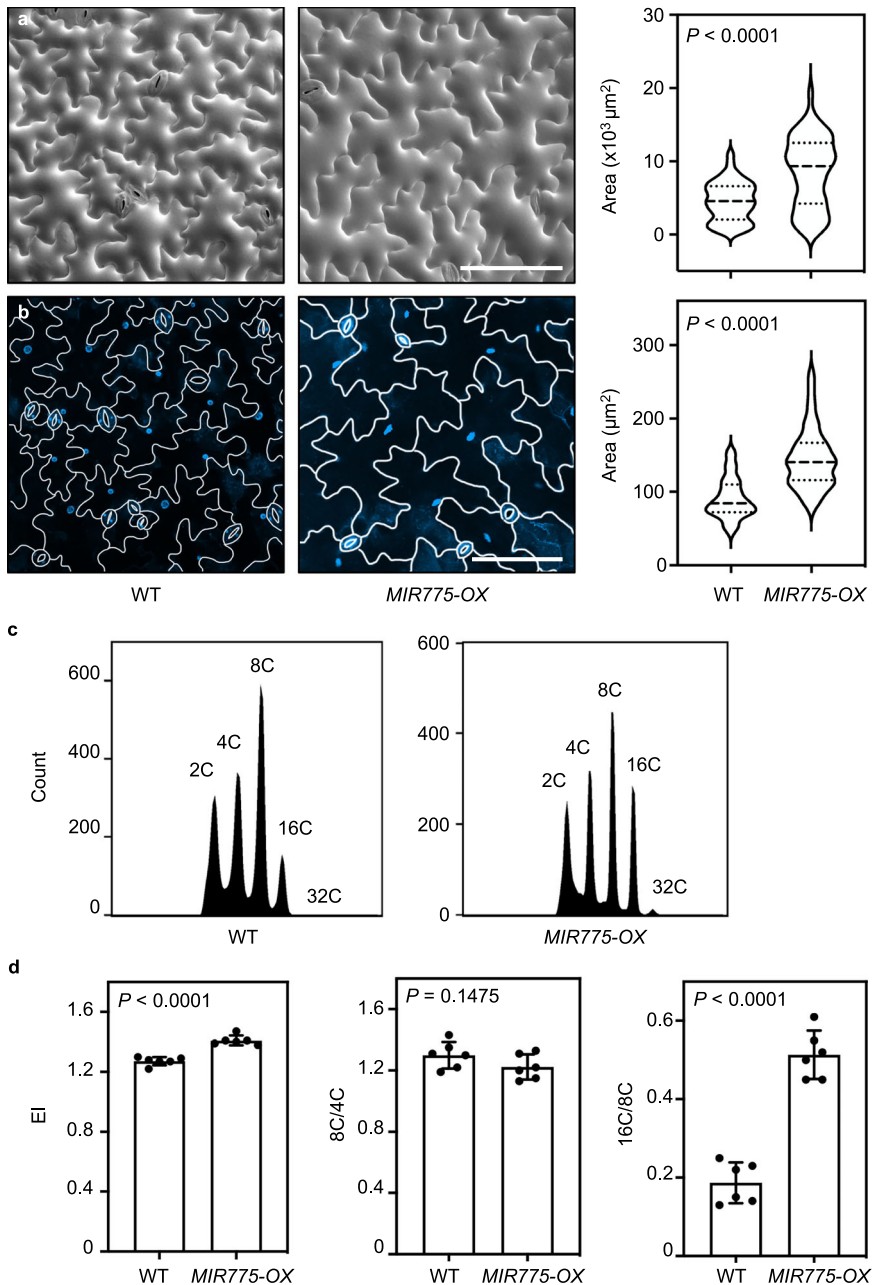

**Fig. 8 | Pectin-regulating miR775 promotes endocycle progression. a, b** Cryo-SEM analysis (**a**) and DAPI staining (**b**) of the adaxial pavement cells in 7-day-old wild type and *MIR775-OX* cotyledons. Scale bars, 100 μm. Quantifications of cell and nucleus sizes are shown on the right. Values are mean ± SD (*n* = 100 in **a** and *n* = 55 in **b**). ***, *p* < 0.001 by Mann-Whitney test. Source data are provided as a Source Data file. **c** Representative ploidy profiles of the third pair of rosette leaves of 4-week-old wild type and *MIR775-OX* plants. **d** Quantification of parameters related to endocycle using ploidy profiles obtained by flow cytometry analysis. Values are mean ± SD (*n* = 6 independent experiments). Statistical analysis was performed using a two-tailed unpaired Student's *t* test. Source data are provided as a Source Data file.

direct assessment of the pectin profiles and wall properties in individual cells at these ploidy levels is not yet available, several lines of evidence provide circumstantial support to this notion. The DNA topoisomerase VI complex is essential for the decatenation of replicated chromosomes during endocycles. It has been reported that the endocycle in mutants defective in various components of the DNA topoisomerase VI complex is stalled at 8 C[9–12]. Interestingly, a reduction in the amount of secreted mucilage, which in Arabidopsis is mainly composed of pectin[10,62], was noticed in the *midget* mutant with defective DNA topoisomerase VI[10], implying that the wall pectin level might have increased in this mutant. It would be interesting to investigate whether pectin level was up-regulated in *midget* and whether

activation of *PG* genes such as *PGL1* might lead to progression of the endocycle beyond the 8C stage in this mutant.

In mutants defective in DNA topoisomerase VI, an extreme dwarfism of plants and a scarcity large cell types that normally endoreduplicate their DNA were exhibited[9–12]. A preliminary survey of the literature revealed that similar morphological phenotypes were reported in many mutants hyper-accumulating cell wall pectin. For example, plants overexpressing *PGX1*, which encodes a PG, displayed enhanced cell elongation in the hypocotyl as a result of reduced abundance and molecular mass of pectin[24]. It has been reported that overexpression of *PECTIN METHYLESTERASE 5* (*PME5*), which promotes HG demethylation, led to larger pavement cells in Arabidopsis

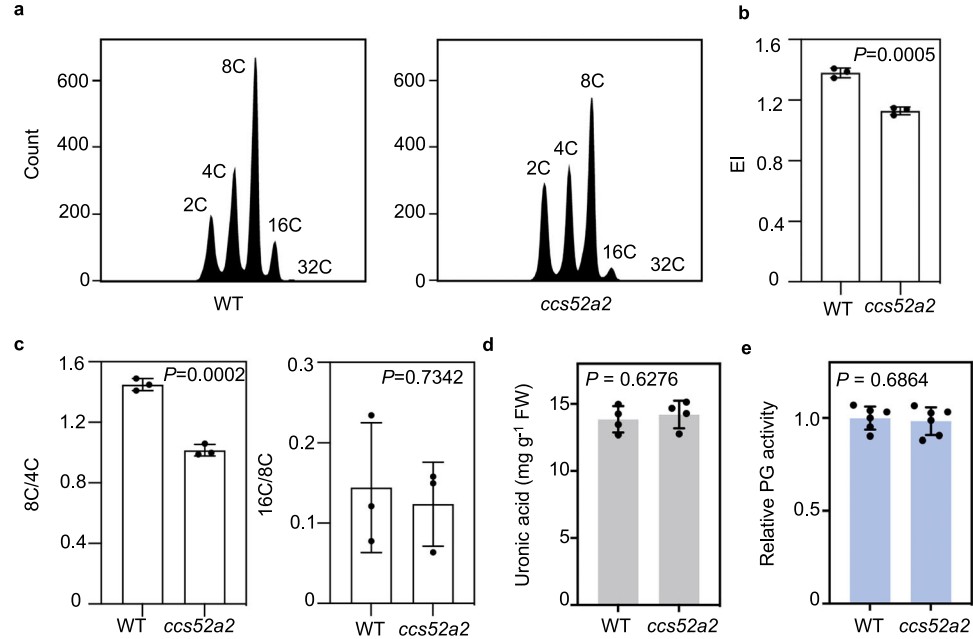

**Fig. 9 | Hindering endocycle onset alone does not impact pectin level.**
**a** Representative ploidy profiles of 7-day-old wild type and *ccs52a2* cotyledons.
**b**, **c** Quantification of endocycle parameters using ploidy profiles obtained by flow cytometry analysis. Values are mean ± SD ($n = 3$ biological replicates). Statistical analysis was performed using a two-tailed unpaired Student's *t* test. Source data are provided as a Source Data file. **d** Comparison of total pectin contents in 7-day-old

wild type and *ccs52a2* seedlings. Values are mean ± SD ($n = 4$ independent experiments). FW fresh weight. Source data are provided as a Source Data file. **e** In vivo PG activity in 7-day-old seedlings. Values are mean ± SD ($n = 6$ independent experiments) of the absorbance normalized to the wild type. Statistical analysis was performed using a two-tailed unpaired Student's *t* test. Source data are provided as a Source Data file.

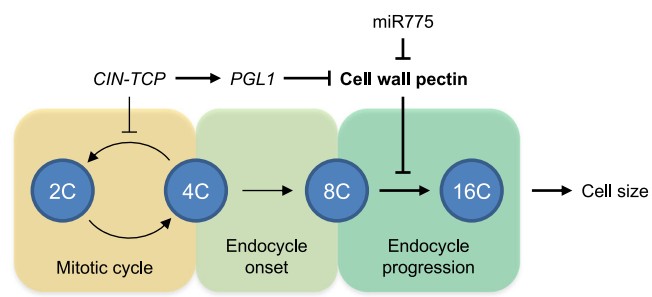

**Fig. 10 | A working model of the role of cell wall pectin in regulating endoreplication and cell size.** Two molecular routes were identified for lifting the restriction of cell wall pectin on endocycle progression. In the first route, the CIN-TCP transcription factors, in addition to promoting endocycle onset by inhibiting the mitotic cell cycle, activate *PGL1* expression to downregulate pectin abundance and polymer sizes to advance endoreduplication beyond the 8 C stage. In the second route, the pectin-downregulating miR775 independently promotes endocycle progression. Successful endocycle progression promotes the enlargement of pavement cells of the leaves.

cotyledons[20]. By contrast, overexpression of *PECTIN METHYLESTERASE INHIBITOR 3* (*PMEI3*), which inhibits HG demethylation, resulted in smaller cotyledon pavement cells[20]. These coincidences should inspire future efforts to test whether the various mutants with altered pectin metabolism would specifically impact endocycle progression.

The finding on the role of CIN-TCP transcription factors in regulating endocycle progression and cell enlargement through PGL1-mediated pectin metabolism adds one more node to the regulatory networks coordinating mechanical forces and nuclear events. This is more conceptually straightforward in animals as the cytoskeleton bridges the nuclear envelope and the plasma membrane, and

mechanical deformations resulted from the extracellular matrix would impact the architecture of the genome[63]. Plants also use the cortical microtubules as the active mechanosensors to transduce mechanical forces[64]. How mechanical forces of the cell wall coordinate the inner bioprocesses to control cell growth is thus a fascinating research direction. On the other hand, mechanical forces could be indirectly perceived by the mechanotransduction pathways, such as wall sensors, mechanosensitive channels, the cytoskeleton, and even the nucleus[65]. For example, some cell wall components could serve as signals for converting mechanical properties into biochemical signals to regulate morphogenesis[66,67]. Future investigation of how plants orchestrate these two important processes would pave the way to coordinately engineer cell wall remodeling and endoreplication to sculpt cell expansion and plant architecture, which determine many critical traits such as biomass accumulation and stress tolerance in crops[4].

## Methods

### Plant materials and growth conditions

All the Arabidopsis (*Arabidopsis thaliana*) materials used in this study were in the Columbia-0 background. The septuple mutant *tcpΔ7* were generated by genetic crossing, and the *35S:MYC-mTCP4* plant were generated by PCR-based mutagenesis[38]. To generate the *PGL1-OX* and *tcpΔ7 PGL1-OX* plants, the coding region of *PGL1* was PCR-amplified from cDNA using primers listed in Supplementary Table 1. The fragment was cloned into the *pJim19* vector, and the resulting construct was transformed into the wild-type and *tcpΔ7* plants, respectively. Transformants were selected on half-strength Murashige and Skoog medium containing 20 mg L$^{-1}$ Hygromycin and allowed to propagate to the T$_2$ generation for subsequent analyses. T-DNA insertion mutants of *ccs52a2* (SALK_073708) and *pgl1-1* (SALK_202104) were obtained from the Arabidopsis Biological Resource Center. The *pgl1-2* line was obtained by the CRISPR/Cas9 method using a pair of sgRNA designed to delete a 1693 bp region in *PGL1* (Supplementary Table 1). Following

transformation of the wild type, $T_1$ transgenic seedlings were genotyped by PCR and sequencing to identify deletion events. A Cas9-free homozygous mutant line was selected at the $T_2$ generation.

To grow Arabidopsis seedlings, seeds were surface-sterilized and plated on agar-solidified half-strength Murashige and Skoog medium. The plates were incubated at 4 °C for three days in the dark and then transferred to a growth chamber with 22 °C/20 °C, 16 h light/8 h dark settings, and a light intensity of 150 μmol m$^{-2}$ s$^{-1}$. Adult plants were maintained in commercial soil and a growth chamber with settings of 22 °C/20 °C, 16 h light/8 h dark cycles, light intensity of 120 μmol m$^{-2}$ s$^{-1}$, and 50% relative humidity. Tobacco (*Nicotiana benthamiana*) plants were maintained in a growth chamber with 25 °C/21 °C, 16 h light/8 h dark settings, a light intensity of 200 μmol m$^{-2}$ s$^{-1}$, and a relative humidity of 50%.

### Flow cytometry
The third pair of rosette leaves of 14 DAS (stage 1), 18 DAS (stage 2), and 28 DAS (stage 3) or 7-day-old cotyledons were prepared for flow cytometry using DNA-selective fluorochromes[68]. Briefly, approximately 250 mg fresh leaves were cut into small pieces in ice-cold Otto I solution (100 mM citric acid, 0.5% (v/v) Tween 20), filtered through a 40 μm nylon mesh (Falcon), and centrifuged at 150 × g for 5 min for rosette leaves or at 100 × g for 5 min for cotyledons. The nuclei were resuspended in 100 μL ice-cold Otto I solution, followed by incubation with 1 mL Otto II solution (400 mM disodium hydrogen phosphate) containing 50 μg mL$^{-1}$ DNase-free RNase (Sigma-Aldrich) and 50 μg mL$^{-1}$ propidium iodide (Sigma-Aldrich). Approximately 15,000 nuclei per sample for rosette leaves or 10,000 nuclei per sample for cotyledons were analyzed using the FACSVerse flow cytometer (Becton Dickinson). Quantitative analysis of ploidy level was conducted using the FlowJo software (v10.4). EI was calculated from the ploidy histograms using the formula: EI = 4 C + 2*8 C + 3*16 C + 4*32 C, where 4 C/8 C/16 C/32 C was the percentage of 4 C/8 C/16 C/32 C nuclei within the sample[39].

### DAPI staining
Ten-day-old cotyledons were stained with 10 μg mL$^{-1}$ DAPI (Sigma-Aldrich) for 20 min in the dark under a vacuum. Nuclear DAPI fluorescence of abaxial epidermal cells was observed using an LSM 800 laser scanning confocal microscope (Zeiss).

### Cryo-SEM
The third pair of rosette leaves or cotyledons or were collected and immediately frozen in subcooled liquid nitrogen (−210 °C). The samples were transferred in vacuum to the cold stage of a PP3010T workstation chamber (Quorum Technologies) for sublimation (−90 °C, 5 min) and platinum sputter coating (10 mA, 30 s). The samples were then transferred to the cold stage in an FEI Helios NanoLab G3 UC scanning electron microscope (Thermo Scientific). Images were acquired using the electron beam at 2 kV and 0.2 nA with a working distance of 10 mm. Cell size was quantitatively measured using ImageJ[69]. The average cell size for each genotype was calculated from a minimum of 100 pavement cells from three individual plants.

### Chemical analysis of cell wall polysaccharides
Cellulose, pectin, and hemicellulose levels in leaves of four-week-old plants were measured using the Cellulose Extraction and Determination kit, the Pectin Extraction and Determination kit, and the Hemicellulose Extraction and Determination kit, respectively, following the manufacturer's protocols (Comin Biotechnology). Following extraction of cell wall polysaccharides, the amount of glucose, uronic acid, and xyloglucan, the primary hydrolysates of cellulose, pectin and hemicellulose[22,70,71], was used to represent contents of the respective polysaccharides. Cellulose and pectin levels were quantified by colorimetry[22,72]. To quantify hemicellulose, approximately 20 mg dried

leaf tissues per sample was homogenized, mixed with 1 mL solution I provided in the kit, heated at 90 °C for 10 min, and centrifuged at 8000 × g for 10 min. The pellets were washed three times in 1 mL distilled water and centrifuged at 8000 × g for 10 min. The pelleted cell wall materials were dried, resuspended in 0.2 mL solution II, and heated at 90 °C for 1 h. After cooled to room temperature, the samples were mixed with 0.2 mL solution III and centrifuged at 8000 × g for 10 min. The hemicellulose content in the supernatants was determined by the DNS method[70] using a NanoDrop 2000c spectrophotometer (Thermo Scientific).

### Immunohistochemistry
Immunofluorescence analysis of cell wall pectin was performed in this study[22,46]. Seven-day-old seedlings were fixed in precooled methanol, vacuum infiltrated for 30 min, and stored overnight at 4 °C. After dehydration, seedlings were embedded in Steedman's wax (Sigma-Aldrich) and cut into 8 μm sections using an RM2255 microtome (Lecia). After rehydration, the sections were pretreated for 1 h with 2% (w/v) BSA in PBS (for the LM19 antibody) or T/Ca/S (20 mM Tris-HCl, pH 8.2, 1 mM CaCl$_2$, 150 mM NaCl, for the 2F4 antibody) buffer. The sections were then incubated overnight with the corresponding primary antibodies diluted 1:500 with 0.1% BSA in PBS or T/Ca/S. After three washes in buffer containing 0.1% (v/v) Tween 20 (Sigma-Aldrich), sections for LM19 (PlantProbes, Cat#LM19; RRID: AB_2734788) and 2F4 (PlantProbes, Cat#PP-2F4) were incubated for 1 h with the secondary antibody Alexa Fluor 546 goat anti-rat IgG (Life Technologies, Cat#A11081) and Alexa Fluor 546 goat anti-mouse IgG (Life Technologies, Cat#A11003) diluted 1:800 with 0.1% BSA in PBS or T/Ca/S, respectively. After additional rinses in buffer containing 0.1% Tween 20, sections were stained with 0.1 mg mL$^{-1}$ Fluorescent Brightener 28 (Sigma-Aldrich) for 10 min. After three washes in water, sections were mounted in ProLong Antifade (Life Technologies) under coverslips. Fluorescence imaging was performed under an LSM 800 laser scanning confocal microscope (Zeiss). Quantitative analysis of fluorescent intensity was performed using ImageJ, with the mean intensity of given areas calculated and normalized using the FB28 signals.

### PG activity assay
Total protein extraction and PG activity assay were performed in this study[24,49]. Briefly, 3 g of seven-day-old seedlings were harvested and ground into fine powder in liquid nitrogen. The samples were homogenized in 5 mL ice-cold extraction buffer (50 mM TrisHCl, pH 7.5, 2.5 mM DTT, 3 mM EDTA, 2 mM PMSF, 1 M NaCl, and 10% (v/v) glycerol) and centrifuged at 18,000 g for 40 min at 4 °C. The supernatant was dialyzed in 50 mM sodium acetate, pH 5.0, using 10,000 MWCO centrifugal filter devices (Millipore). PG activity was assessed as the increase in reducing end groups using 0.2% (w/v) polygalacturonic acid (Sigma-Aldrich) as the substrate and measured at 276 nm absorbance on a NanoDrop 2000c spectrophotometer.

### FPLC analysis of pectin molecular sizes
Seven-day-old seedlings were harvested and ground into fine powder in liquid nitrogen. The powder was washed with 70% ethanol after treatment with 1:1 (v/v) chloroform:methanol. The air-dried cell wall residues were used for pectin extraction using 1,2-cyclohexylenedinitrilotetraacetic acid (CDTA)[21,24]. Five mg CDTA-soluble lyophilized pectin powder was dissolved in 1 mL 0.1 M sodium acetate and filtered through a 0.22 μm filter. A Superdex 75 10/300 GL gel filtration column (GE Healthcare) with gel media equilibrated with 0.1 M sodium acetate, pH 5.0, was used to fractionate the CDTA-soluble polymers. The sample was eluted at a flow rate of 500 μL per min, and the MWGF200 Size Standard Kit (SigmaAldrich) was used to calibrate molecular mass. Fifty fractions with 500 μL each were collected and analyzed for uronic acid content[21]. Briefly, 250 μL of each fraction was mixed with 1 mL 0.0125 M sodium tetraborate

(SigmaAldrich) in sulfuric acid solution, boiled for 5 min, and cooled to room temperature. Absorbance of the sample was measured at 520 nm as the background. The sample was then mixed with 20 μL of 0.15% (w/v) mhydroxydiphenyl (SigmaAldrich), dissolved in 0.5% (w/v) sodium hydroxide, and incubated at room temperature for 10 min. The content of uronic acid was measured as the absorbance at 520 nm.

## RNA-seq analysis

RNA samples were collected from young leaves of four-week-old wild-type and *tcpΔ7* plants and sent to Beijing Genomics Institute (China) for sequencing. Trimmomatic was used to remove low-quality reads with the default parameters and to retain reads longer than 50 bases. The clean reads were mapped to the reference genome (TAIR10_Araport11) using HISAT (v2.1.0). Transcript quantification was processed by StringTie (v1.0.4). DEseq2 was implemented to identify differentially expressed genes with Qvalue (adjusted *P*-value) < 0.05. The false discovery rate (FDR) method was applied for correcting the *P*-value.

AgriGO 2.0 was used for GO term enrichment analysis[72]. For differentially expressed genes, we estimated GO term enrichment in the biological process category using the Fisher test. GO terms in the biological process category was clustered and analyzed using the simplifyEnrichment package[42]. Similarity between each pair of GO terms was calculated with k-means. GO terms with similarity >0.6 were used to construct an association network.

## Quantitative transcript analysis

Total RNA was extracted using the Quick RNA Isolation Kit (Huayueyang). First strand cDNA synthesis and qPCR were performed respectively with the Fasting RT Kit with gDNase (Tiangen) and the SuperReal Premix Plus SYBR Green (Tiangen) on a QuantStudio 3 Real-Time PCR System (Applied Biosystems) in standard mode. *ACTIN7* was used as an internal control. Relative transcript abundance was calculated using the comparative threshold cycle method.

## ChIP-qPCR

Approximately 2 g of seven-day-old wild-type and MYC-mTCP4 seedlings[38] were harvested and crosslinked in 1% formaldehyde for 30 min. The ChIP procedure was performed following the standard protocol[73]. AntiMYC Agarose Affinity Gel antibody (Sigma-Aldrich, Cat#A7470) was used to perform the immunoprecipitation. IgG was used as the negative control for the MYC antibody. Following ChIP, qPCR was performed to examine the enrichment level of selected DNA fragments using primers listed in Supplementary Table 1. The enrichment of each target site was calculated as the percentage of coimmunoprecipitated DNA relative to the corresponding input DNA.

## Promoter activity assays

A 3000 bp promoter sequence upstream of the predicted translational initiation site of *PGL1* was PCR amplified from genomic DNA using primers listed in Supplementary Table 1. The fragment was cloned into the *p1305.1* vector to generate the *pPGL1:LUC-35S:REN* reporter construct. The *pPGL1^m^:LUC-35S:REN* construct with the nucleotides of the three CIN-TCP binding sites (site II, CGGNCC-I and CGGNCC-II) mutated was generated using primers listed in Supplementary Table 1. The coding region of *TCP4* was cloned from cDNA using primers listed in Supplementary Table 1 and inserted into the *pJim19* vector to generate the *35S:TCP4* effector construct. The reporter constructs, combined with either the *pJim19* empty vector or the *35S:TCP4* effector, were used to transform *Agrobacterium* cells, which was prepared to OD$_{600}$ of 1.0 in the infiltration buffer[74]. The *Agrobacterium* cells were used to infiltrate tobacco leaf epidermal cells. After grown at 24 °C for 2 days, the infiltrated leaves were detached, sprayed with 20 g L$^{-1}$ potassium luciferin (Gold Biotechnology), and incubated in darkness for 5 min before imaging. Luminescence imaging was performed with the Lumazone system (Roper Technologies) with 5 min exposure time and

1 MHz ADC rate. Quantification of luminescence was carried out with ImageJ using integrated density.

## BLI assay

The binding kinetics of TCP2 and TCP3 to the *PGL1* promoter was monitored using a GatorPrime instrument (Gator Bio) with minor modifications[54]. In brief, the trigger factor- and His-tagged recombinant TCP2 and TCP3 based on the pCold TF DNA vector (Takara Bio) were purified from *E. coli* strain Rossetta DE3. The biotin-labeled DNA fragments containing the binding motifs (site II, CGGNCC-I and CGGNCC-II) or mutated versions (site II^m^, CGGNCC-I^m^ and CGGNCC-II^m^) were loaded onto the streptavidin-coated sensors (Gator Bio, 160002) from a solution of 100 nM. After immobilization, the sensors were dipped into various concentrations of TCP2 (70 to 1200 nM) or TCP3 (112.5 to 1800 nM) for 120 s of association and 180 s of dissociation in a PBST buffer supplied with 0.2% (w/v) BSA. The processed data curves for the association and dissociation steps were used to calculate the $K_{off}$, $K_{on}$, and $K_d$ values with a 1:1 binding model (global fit) using the GatorOne software (Gator Bio).

## Statistical analysis

Statistical analyses were performed using Prism 8 (v8.4.0). For the two-sample comparison, Student's *t* test was performed if the sample data conforms to Gaussian distribution, if not, the Mann-Whitney test was performed. For the multiple-sample comparison, one-way ANOVA was performed if the sample data conforms to a Gaussian distribution, if not, the Kruskal-Wallis test was performed. The number of samples and statistical significance were indicated in the relevant figure legends.

## Reporting summary

Further information on research design is available in the Nature Portfolio Reporting Summary linked to this article.

## Data availability

The RNA-seq data generated in this study have been deposited in the National Center for Biotechnology Information (NCBI) Database with the BioProject ID PRJNA971251. Source data are provided in this paper.

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

## Acknowledgements

We thank Dr. Guilan Li at the National Center for Protein Science at Peking University, Jun Hu at the Core Facilities of School of Life Sciences, Peking University, and Dr. Guoqiang Wang for technical assistance in FPLC operation and data analysis, Cryo-SEM operation, and BLI analysis, respectively. This work was supported by grants from the National Natural Science Foundation of China (32370368) and the Taishan Scholars Program to L.L.

## Author contributions

L.L. conceived this study. F.S., H.Z., M.W., D.Z., Y.Y., and Z.L. carried out the experiments. F.S., Z.K., and L.X. analyzed the data. G.Q. provided critical experimental materials. F.S., H.Z., and L.L. wrote the manuscript.

## Competing interests

The authors declare no competing interests.
