## [Transparent Peer Review file · Nature Communications]

The CIN-TCP transcription factors regulate endocycle progression and pavement cell size by promoting cell wall pectin degradation

Corresponding Author: Professor Lei Li

Version 0:

Reviewer comments:

Reviewer #1

(Remarks to the Author)

CIN-TCP transcription factors are well-described regulators of cell cycle exit during leaf development. They operate in a dose dependent manner: the more family members are mutated, the more severe the phenotype. Within this work a septuple CIN-TCP mutant (*tcpd7*) has been generated that shows a cell division phenotype being strongly related to that of the previously reported pentuple mutant, i.e. a late cell cycle exit that is exemplified by smaller cells with a reduced cellular ploidy level due to a delayed onset of endoreduplication (compared to the control plants). Transcriptome analysis on these plants was used to pinpoint the pectin metabolic gene PGL1 as a CIN-TCP target that through control of pectin abundance is put forward as a CIN-TCP dependent regulator of cell size and ploidy level. A similar role is attributed to a miRNA targeting another pectin biosynthetic gene.

Although I appreciate the idea of CIN-TCPs controlling cell cycle exit and endocycle onset through the activation of pectin modifying genes, I do not see any convincing evidence for this model in the given data. A major problem here is that the growth phenotype of the *tcpd7* mutant versus control plants is very outspoken. This extreme difference is reflected in the outspoken change in the transcriptome with > 4.600 genes (thus > 20% of the complete transcriptome) being differentially expressed. Likewise, the TCPs are very promiscuous binding factors, with e.g. the TCP4 factor binding on its own already about 800 genes. Combined this making it very difficult to pinpoint direct key CIN-TCP target genes. And although I appreciate the PGL1 transgenic data, I see little evidence for this gene to control endocycle progression in a TCP-dependent manner. To achieve this result, at least the following experiments should be conducted:

- Generate a PGL1 reporter and study its activity during the process of leaf differentiation in a wild type versus *tcpd7* mutant background (I guess that this can be done as well in a lower order mutant). This should be complemented by a reported line in which the putative CIN-TCP binding site is mutated.
- The executed *tcpd7* x PGL1-OE experiment is not the correct one to proof the hypotheses, as both have an opposite phenotype. What you see in the double transgenic line might be simply the sum of the two independent phenotypes. The authors should rather introduce the *pgl1* mutation in the *tcpd7* (or lower order) mutant background to demonstrate epistasis.

Additionally major comments:

Immunolabeling experiment are rarely quantitative (slide variation). Moreover, due to the huge difference in cell density, the fluorescence signal will appear more intense in the sample with smallest cells. This might be remediated mounting both genotypes on the same slide and by quantifying the immuno signals at the cellular level.

For the *pgl1* knockout data on an independent line should be provided. Alternatively, the authors should perform a complementation experiment.

The *pgl1* KO shows a reduced 16C population, whereas the overexpression line shows an increase. This fits the hypothesis. However, the differences observed are in line with the variation between wild type samples shown in Figure 5a versus Figure 6e (both showing data on the 3rd leaf of 4 week-old old plants), questioning the significance of the observed ploidy changes. Rather than focusing on one (late) timepoint, it would be better to map the change in ploidy in the leaf over

time.

Reviewer #2

(Remarks to the Author)

Plant cells are often endoreplicated, and the degree of endoreplication is often, but not always coupled with increased cell size. These two processes play important roles in the generation of plant form. However, while we have learned a great deal about the cell cycle machinery responsible for initiating and maintaining endoreplication, we know little about the machinery responsible for coordinating endoreplication and cell size. This manuscript adds an important new piece to this puzzle, clearly demonstrating that constraints on cell expansion via cell wall pectin have a feedback effect on endoreplication.

This is one of the most logically constructed and clearly written manuscripts I have ever read. Starting from gene expression effects resulting from a septuple CIN-TCP transcription factor mutant that reduces both endoreplication and cell size in cotyledons, the authors identify PGL1, a gene encoding a polygalacturonase, as a direct target upregulated by these TCPs. They show that the mutant has effects on wall pectin levels consistent with the altered gene expression, and show that over-expression of PGL1 promotes expansion and endoreplication and partially rescues the tcp mutant, while a pgl1 loss-of-function mutant reduces both cell size and endoreplication. They also show that over-expression of a microRNA that regulates pectins independently of PGL1 has effects consistent with regulation of endoreplication and cell size by pectins. Finally, they show that a ccs52 mutant that is part of the cell cycle machinery of endoreplication has no effect on pectin content of the wall, convincingly showing that pectin content itself has a feedback effect on the endoreplication cycle.

This paper convincingly demonstrates all of its main points, and I have only a few major points, none of which are essential to the points that the authors are trying to make:

1. Does PGL1 overexpression rescue or partially rescue the ccs52a2 mutant endoreplication defect? This is probably the easiest of my suggestions to do.
2. The authors convincingly show that CIN-TCPs bind to and directly regulate PGL1 expression. It would be interesting to know whether mutating the putative binding sites in the PGL1 promoter prevented activation of PGL1 in the transient expression system, but this is not necessary for the conclusions presented.
3. The authors have very convincingly shown that cell wall biochemistry directly feeds back on the cell cycle to increase endoreplication. This is a major piece that we have been missing from the puzzle of how endoreplication and cell size are coordinated. But showing that cell wall changes associated with cell expansion influence endoreplication leaves us with another puzzle: How does the cell wall regulate the endocycle machinery? I do not expect the authors to do any further experiments to address this question. But would like the authors to acknowledge this question in the Discussion, and I would encourage them to speculate if they can, and if possible refer to examples where mechanical forces acting on a cell influence transcription and nuclear events. I believe that the Hamant lab has some work that speaks to this, and there is probably more. I feel that this would make the manuscript stronger, and of greater general interest.

Minor points (two typos):

4. Line 301 "was significantly increased than tcpd7" something is wrong here.
5. Line 369 "resorted" should be restored, I think.

Version 1:

Reviewer comments:

Reviewer #1

(Remarks to the Author)

I would like to express my appreciation to the authors for their handling of a part of the previous comments. They have done a good job in providing more evidence that PGL1 could be a direct target of TCP and providing an independent knockout line. I also appreciate the clear link between PGL1 activity, pectin content and cell size, demonstrated by independent experiments.

What I'm still not convinced of is that the cell size phenotypes are due to effects at the ploidy level, especially since the effects on cell size are very pronounced, whereas those on ploidy are minimal at best (with only minor differences in 16C ploidy cells, which represent the minority of the cell population). In my opinion, these are only secondary effects on the timing of cell cycle exit. I hinted at this in my previous review by asking for changes in ploidy levels during leaf development to be mapped. In response, the author included a 4-week versus 5-week comparison in Figure 5, but this was obviously not the idea, as these are both late time points.

To clarify (and expand on the comment):

1. Ploidy analysis should be carried out at different developmental stages during leaf development, comparing a stage where leaves are still dividing, in the process of exiting the cell cycle, and a stage where leaves are mature.

2. In addition, the authors should give data not only on cell size but also on cell number. Such data can be obtained by also measuring leaf (or cotyledon) size and estimating the total epidermal cell number for these data by extrapolation.

My prediction from these analyses (minimal to be done on the *tcp7* and *pgl1* mutants) is that the time of cell cycle exit would be delayed, which would explain the smaller cells (rather than being a consequence of an altered endocycle). If the authors find no differences in total cell number, this will support their own hypothesis. I would like to emphasize that showing an effect of pectin on the timing of cell cycle exit (rather than on endocycle directly) and thus on cell size, would still be a nice result. On the other hand, publishing the paper in its current form could potentially propagate the false hypothesis that pectin abundance drives the endocycle.

Additional comments:

1. It is mentioned that the 4C/8C ratio is a surrogate for endocycle onset. This is true when young leaves are used, but not when ploidy measurements are made on 4-week-old plants (as is the case in this paper). As mentioned above, the best way to demonstrate a clear effect on ploidy levels is to measure ploidy levels during the process of leaf development, from a young dividing leaf to a mature leaf.
2. Often, data from different tissues or different ages are compared. This is the case for many of the experiments, but to give an example: Figure 1 measures cell size in 7-day-old cotyledons and correlates this with ploidy measurements on 4-week-old rosettes. The most problematic case is the last figure, where ploidy measurements are made on 4-week-old *ccs52a* plants, but pectin measurements are made on 7-day-old seedlings. Using these data to conclude that there is no relationship between ploidy and pectin content is incorrect. Pectin should be measured at the same time and in the same tissue as when ploidy differences are observed.
3. Line 186: The plates are incorrectly referenced.

Reviewer #2

(Remarks to the Author)

1. The authors have done pretty much everything that both reviewers have asked for, which is commendable. In particular, with the addition of showing that mutation of TCP binding sites in the PGL1 promoter abolishes regulation of PGL1 by several TCPs convincingly closes the loop demonstrating that TCPs regulate PGL1 expression.
2. I very much agree with the authors that Reviewer 1's comment on the *tcpd7* PGL1-OX experiment. This IS the correct epistasis experiment, which can only be convincingly done with mutants with opposing, or at least substantially different, phenotypes. This experiment, as done in the original manuscript, clearly demonstrated that the effect of PGL1 expression in promoting endoreplication is independent of the TCPs, which was the whole point. As they predicted, nothing interesting comes from the *pgl1 tcp-delta3* mutant line with regard to the endoreplication phenotype relevant to the manuscript, which is what I would have predicted as well. And while the multiple mutant has the curled leaf phenotype of the *tcp-delta3* mutant on its own, showing epistasis of this aspect of the phenotype, this is precisely due to the fact noted by reviewer 1 that TCPs regulate many genes other genes. This is not surprising, and in no way contradicts the author's conclusions. But they went the extra mile and did the experiment.
3. The observation in Fig. R2 in the review showing that PGL1-OX can suppress that endoreplication defect of the *ccs52* mutant is exciting, and highlights the importance of the work. The authors clearly would like to hang onto this result and build a new story around it. But it shows that this work is likely to be part of a growing story with continuing important impact.

Version 2:

Reviewer comments:

Reviewer #1

(Remarks to the Author)

I'm OK with this second round of revisions

Responses to Comments by Reviewer #1

CIN-TCP transcription factors are well-described regulators of cell cycle exit during leaf development. They operate in a dose dependent manner: the more family members are mutated, the more severe the phenotype. Within this work a septuple CIN-TCP mutant (tcpd7) has been generated that shows a cell division phenotype being strongly related to that of the previously reported pentuple mutant, i.e. a late cell cycle exit that is exemplified by smaller cells with a reduced cellular ploidy level due to a delayed onset of endoreduplication (compared to the control plants). Transcriptome analysis on these plants was used to pinpoint the pectin metabolic gene PGL1 as a CIN-TCP target that through control of pectin abundance is put forward as a CIN-TCP dependent regulator of cell size and ploidy level. A similar role is attributed to a miRNA targeting another pectin biosynthetic gene.

Although I appreciate the idea of CIN-TCPs controlling cell cycle exit and endocycle onset through the activation of pectin modifying genes, I do not see any convincing evidence for this model in the given data. A major problem here is that the growth phenotype of the tcpd7 mutant versus control plants is very outspoken. This extreme difference is reflected in the outspoken change in the transcriptome with > 4,600 genes (thus > 20% of the complete transcriptome) being differentially expressed. Likewise, the TCPs are very promiscuous binding factors, with e.g. the TCP4 factor binding on its own already about 800 genes. Combined this making it very difficult to pinpoint direct key CIN-TCP target genes. And although I appreciate the PGL1 transgenic data, I see little evidence for this gene to control endocycle progression in a TCP-dependent manner. To achieve this result, at least the following experiments should be conducted:

Generate a PGL1 reporter and study its activity during the process of leaf differentiation in a wild type versus tcpd7 mutant background (I guess that this can be done as well in a lower order mutant). This should be complemented by a reported line in which the putative CIN-TCP binding site is mutated.

Response: Your concern on the direct regulation of *PGL1* by CIN-TCPs in regulating endocycle is well received, given that TCP proteins extensively regulate gene expression. That is exactly why we had gone great lengths to combine RNA-seq, GO analysis, molecular assays, biochemical analyses and genetic interactions to pinpoint *PGL1* as a target for CIN-TCPs in endocycle regulation.

Regarding evidence supporting CIN-TCPs activation of *PGL1* transcription via binding to its promoter, we had already presented several lines of evidence, including RNA-seq, RT-qPCR, mTCP4 ChIP, and promoter activity assays in the original manuscript (Fig. 3). As suggested, we further confirmed binding of CIN-TCPs to the *PGL1* promoter by mutating the three CIN-TCP binding motifs (site II, CGGNCC-I and CGGNCC-II). In the transient expression experiment, we generated the *pPGL1^m:LUC-35S:REN* dual reporter construct with the three binding sites mutated. We examined the LUC/REN ratio by co-infiltration with the *35S:TCP2*, *35S:TCP3*, *35S:TCP4*, and *35S:TCP17* constructs in tobacco leaf epidermal cells. We showed that the LUC/REN ratios were all markedly decreased comparing to the original *PGL1* promoter. These results were added to Fig. 3 and Supplementary Fig. 5 in the revised manuscript.

Moreover, we performed Biolayer Interferometry (BLI) assay to measure the binding kinetics between TCP2 or TCP3 and the three binding motifs in the *PGL1* promoter. In this assay, the dissociation constant of the binding could be calculated based on the protein and DNA fragment concentrations. We found that all the combinations between TCP2 or TCP3 and the DNA motifs showed high binding affinities with the dissociation constant consistently in the $\sim 0.1 \mu\text{M}$ range. In contrast, mutating these binding sites abolished the binding by TCP2 or TCP3. These results were added as Supplementary Fig. 6 in the revised manuscript.

Together the newly added results corroborate our conclusion that CIN-TCP2 regulate *PGL1* via binding to the specific DNA motifs in the *PGL1* promoter.

The executed tcpd7 x PGL1-OE experiment is not the correct one to proof the hypotheses, as both have an opposite phenotype. What you see in the double transgenic line might be simply the sum of the two independent phenotypes. The authors should rather introduce the pgl1 mutation in the tcpd7 (or lower order) mutant background to demonstrate epistasis.

Response: Actually, the overarching finding in this study is that CIN-TCPs activate *PGL1* to reduce pectin level for endocycle progression. In fact, *PGL1* transcript abundance decreased drastically in *tcpΔ7* (Fig. 3a, b). Therefore, we generated the *tcpΔ7 PGL1-OX* lines to investigate whether *PGL1* could rescue the defects in *tcpΔ7* regarding pectin level and endocycle progression. We showed that ***PGL1* overexpression indeed rescued the endocycle progression defect** (Fig. 6) and **restored cell wall pectin accumulation in *tcpΔ7*** (Fig. 7). We consider these results as critical evidence for the proposed model that CIN-TCPs activate *PGL1* to reduce pectin level for endocycle progression (Fig. 10).

On the contrary, mutating *PGL1* in the *tcp* background is **NOT** expected to produce phenotypic changes regarding endocycle progression as *PGL1* transcript is virtually not detected (Fig. 3b). Nonetheless, we knocked out *PGL1* in *tcpΔ3* (*tcp3/4/10*) using CRISPR-Cas9 as suggested. We found that the already very low expression level of *PGL1* in *tcpΔ3* was indeed further declined in *tcpΔ3 pgl1* (see Fig. R1a below). As expected, we did not observe noticeable changes in leaf morphology (Fig. R1b) and endocycle progression (Fig. R1c and d) when comparing *tcpΔ3 pgl1* to *tcpΔ3*.

Overall, the genetic results demonstrate that *PGL1* is an important downstream target for CIN-TCPs in the regulation of endocycle progression.

Fig R1. Characterization of *tcpΔ3 pgl1*. **a** Relative *PGL1* transcript abundance in the cotyledons of the indicated genotypes. The expression level was determined by RT-qPCR and normalized to the wild type. Values are mean \pm SD ($n = 3$). Different letters indicate genotypes with significant differences ($p < 0.01$ by one-way ANOVA). **b** Morphology of representative 8-day-old seedlings of the indicated genotypes. Scale bar, 0.5 cm. **c** Representative ploidy profiles of the third pair of leaves of 4-week-old plants using flow cytometry. **d** Quantification of parameters related to endocycle. Values are mean \pm SD ($n = 5$). Different letters indicate genotypes with significant differences ($p < 0.05$ by one-way ANOVA).

Immunolabeling experiment are rarely quantitative (slide variation). Moreover, due to the huge difference in cell density, the fluorescence signal will appear more intense in the sample with smallest cells. This might be remediated mounting both genotypes on the same slide and by quantifying the immuno signals at the cellular level.

Response: Thank you for the advice. In this experiment, care was taken to uniformly section the samples and the quantitative results were normalized by the FB28 signals at the cellular level. The Methods section was revised to emphasize these points (lines 525 to 526). Furthermore, we

performed chemical quantification as well as FPLC analysis of cell wall polysaccharides to corroborate the immunolabeling results.

*For the *pgl1* knockout data on an independent line should be provided. Alternatively, the authors should perform a complementation experiment.*

Response: As suggested, we employed the CRISPR/Cas9 system to delete the entire coding region of *PGL1* to generate an independent *pgl1* allele (please see Supplementary Fig. 6 in the revised manuscript). In the homozygous deletion lines (named *pgl1-2*), *PGL1* expression was clearly compromised and the leaves were much smaller than the wild type. These phenotypes were consistent with those of the T-DNA insertion line, which was named *pgl1-1* in the revised manuscript.

*The *pgl1* KO shows a reduced 16C population, whereas the overexpression line shows an increase. This fits the hypothesis. However, the differences observed are in line with the variation between wild type samples shown in Figure 5a versus Figure 6e (both showing data on the 3rd leaf of 4 week-old old plants), questioning the significance of the observed ploidy changes. Rather than focusing on one (late) timepoint, it would be better to map the change in ploidy in the leaf over time.*

Response: Thank you for the advice. As suggested, we performed flow cytometry analysis to map the ploidy changes in the wild type, *pgl1* and *PGL1-OX* rosette leaves over two development stages. In general, the larger cells in 5-week-old leaves showed a higher ploidy level than those of the 4-week-old leaves. The population of 16C nuclei was remarkably decreased in both 4-week-old and 5-week-old *pgl1* leaves compared to the wild type. On the contrary, *PGL1-OX* showed an increased 16C population at both stages. Together with the calculated EI that was significantly lower in *pgl1* but higher in *PGL1-OX* (Fig. 5b), these results indicate that PGL1 promotes endoreplication over leaf development.

To further discern whether PGL1 impacts endocycle onset or progression, we calculated the 8C/4C and 16C/8C ratios. At both stages, we found that the 8C/4C ratios showed no significant difference between *pgl1* or *PGL1-OX* against the wild type (Fig. 5c). On the contrary, the 16C/8C ratio was significantly decreased in *pgl1* but significantly increased in *PGL1-OX* relative to the wild type at both ages (Fig. 5d). These results demonstrated that PGL1 positively regulates endocycle progression over leaf development and were added as Fig. 5 in the revised manuscript. The text was revised accordingly (lines 261 to 287).

Responses to Comments by Reviewer #2

Plant cells are often endoreplicated, and the degree of endoreplication is often, but not always coupled with increased cell size. These two processes play important roles in the generation of plant form. However, while we have learned a great deal about the cell cycle machinery responsible for initiating and maintaining endoreplication, we know little about the machinery responsible for coordinating endoreplication and cell size. This manuscript adds an important new piece to this puzzle, clearly demonstrating that constraints on cell expansion via cell wall pectin have a feedback effect on endoreplication.

This is one of the most logically constructed and clearly written manuscripts I have ever read. Starting from gene expression effects resulting from a septuple CIN-TCP transcription factor mutant that reduces both endoreplication and cell size in cotyledons, the authors identify PGL1, a gene encoding a polygalacturonase, as a direct target upregulated by these TCPs. They show that the mutant has effects on wall pectin levels consistent with the altered gene expression, and show that over-expression of PGL1 promotes expansion and endoreplication and partially rescues the tcp mutant, while a *pgl1* loss-of-function mutant reduces both cell size and endoreplication. They also show that over-expression of a microRNA that regulates pectins independently of PGL1 has effects consistent with regulation of endoreplication and cell size by pectins. Finally, they show that a *ccs52* mutant that is part of the cell cycle machinery of endoreplication has no effect on pectin content of the wall, convincingly showing that pectin content itself has a feedback effect on the endoreplication cycle.

This paper convincingly demonstrates all of its main points, and I have only a few major points, none of which are essential to the points that the authors are trying to make:

1. Does PGL1 overexpression rescue or partially rescue the ccs52a2 mutant endoreplication defect? This is probably the easiest of my suggestions to do.

Response: Thank you for the encouraging assessment of our work and for this constructive advice. We agree that it is a superb idea to test whether PGL1 still exerts its role when the endocycle is directly blocked in mutants such as *ccs52a2*. As suggested, we generated the *ccs52a2 PGL1-OX* line by overexpressing *PGL1* in *ccs52a2* (please see Fig. R2a and b below). As previously reported (Baloban et al., 2013), *ccs52a2* leaves showed drastically reduced sizes in comparison to the wild type, which was effectively rescued in *ccs52a2 PGL1-OX* (Fig. R2a). Interestingly, the significantly reduced endoreplication index (EI) in *ccs52a2* was recovered to the same level as *PGL1-OX* (Fig. R2c and d). These results suggest that *PGL1* overexpression could at least partially rescue the endoreplication defect of *ccs52a2*. Given this intriguing observation, further studies are needed to fully clarify the relationship between CCS52A-mediated endocycle and PGL1-mediated cell wall remodeling. Therefore, we hope to report this finding in a future story with more comprehensive data.

Fig R2. Characterization of *ccs52a2* PGL1-OX. **a** Morphology of representative 8-day-old seedlings of the indicated genotypes. Scale bar, 2 mm. **b** Relative *PGL1* transcript abundance determined by RT-qPCR in the cotyledons of the indicated genotypes. Values are mean \pm SD ($n = 3$) normalized to the wild type. **c** Representative ploidy profiles of the third pair of leaves of 4-week-old plants. **d** Quantification of endoreplication index (EI). Values are mean \pm SD ($n = 3$). Different letters indicate genotypes with significant differences ($p < 0.05$ by one-way ANOVA).

2. The authors convincingly show that CIN-TCPs bind to and directly regulate *PGL1* expression. It would be interesting to know whether mutating the putative binding sites in the *PGL1* promoter prevented activation of *PGL1* in the transient expression system, but this is not necessary for the conclusions presented.

Response: Thank you for the constructive suggestion. There are three CIN-TCP binding sites in the *PGL1* promoter, namely, site II, CGGNCC-I and CGGNCC-II. As suggested, we generated the *pPGL1^m:LUC-35S:REN* dual reporter construct by mutating the nucleotides of the three binding sites. We found that the ratio of LUC/REN significantly declined in comparison to that of *pPGL1:LUC-35S:REN* when combined with transiently expressed TCP2/3/417, indicating that the CIN-TCPs regulate the *PGL1* promoter via the binding motifs. These results were added to the revised manuscript as Fig. 3 and Supplementary Fig. 5.

Moreover, we performed Bi-layer Interferometry assay to measure the binding kinetics between TCP2 or TCP3 and the three DNA motifs. In this assay, the dissociation constant of the binding could be calculated based on the protein and DNA fragment concentrations. We found that both TCP2 and TCP3 showed high binding affinities to DNA motifs with the dissociation constants in the $\sim 0.1 \mu\text{M}$ range. In contrast, mutating these binding sites abolished the binding by TCP2 or TCP3. These results were added as Supplementary Fig. 6 in the revised manuscript.

3. The authors have very convincingly shown that cell wall biochemistry directly feeds back on the cell cycle to increase endoreplication. This is a major piece that we have been missing from the puzzle of how endoreplication and cell size are coordinated. But showing that cell wall changes associated with cell expansion influence endoreplication leaves us with another puzzle: How does the cell wall regulate the endocycle machinery? I do not expect the authors to do any further experiments to address this question. But would like the authors to acknowledge this question in the Discussion, and I would encourage them to speculate if they can, and if possible, refer to examples where mechanical forces acting on a cell influence transcription and nuclear events. I believe that the Hamant lab has some work that speaks to this, and there is probably more. I feel that this would make the manuscript stronger, and of greater general interest.

Response: Thank you for this insightful suggestion. We agree that a major implication of this work is that pectin-mediated cell wall remodeling regulates the endocycle machinery. Actually, in a preliminary atomic force microscopy (AFM) study, we found that the elastic modulus of the epidermal cells of *tcpΔ7* was higher than the wild type (please see Fig. R3 below). That is to say, the smaller epidermal cells in *tcpΔ7* were more rigid than the wild type, indicating that mechanical properties are coupled with endocycle. Thus, a very interesting and important question awaits to be addressed is how does mechanical forces influence the endocycle machinery in the nucleus. As suggested, we added a discussion on this question in the revised manuscript (lines 419 to 432).

Fig. R3 The *tcpΔ7* epidermal cells were more rigid than the wild type. Individual *tcpΔ7* and wild type cotyledon epidermal cells were mapped by AFM. Shown are three-dimensional cell topography overlaid with cell wall elastic moduli, which are pseudo-colored according to the color scheme shown on the left.

4. Line 301 “was significantly increased than tcpd7” something is wrong here.

Response: This sentence was modified in the revised manuscript (now line 315) to more clearly state the result: “relative PG activity in *tcpΔ7 PGLI-OX* was significantly higher than that in *tcpΔ7*.”

5. Line 369 “*resorted*” should be restored, I think.

Response: Thank you for pointing out this error, which was fixed in the revised text (now line 383).

Responses to Comments by Reviewer #1

I would like to express my appreciation to the authors for their handling of a part of the previous comments. They have done a good job in providing more evidence that PGL1 could be a direct target of TCP and providing an independent knockout line. I also appreciate the clear link between PGL1 activity, pectin content and cell size, demonstrated by independent experiments.

What I'm still not convinced of is that the cell size phenotypes are due to effects at the ploidy level, especially since the effects on cell size are very pronounced, whereas those on ploidy are minimal at best (with only minor differences in 16C ploidy cells, which represent the minority of the cell population). In my opinion, these are only secondary effects on the timing of cell cycle exit. I hinted at this in my previous review by asking for changes in ploidy levels during leaf development to be mapped. In response, the author included a 4-week versus 5-week comparison in Figure 5, but this was obviously not the idea, as these are both late time points.

To clarify (and expand on the comment):

1. Ploidy analysis should be carried out at different developmental stages during leaf development, comparing a stage where leaves are still dividing, in the process of exiting the cell cycle, and a stage where leaves are mature.

2. In addition, the authors should give data not only on cell size but also on cell number. Such data can be obtained by also measuring leaf (or cotyledon) size and estimating the total epidermal cell number for these data by extrapolation.

My prediction from these analyses (minimal to be done on the tcp7 and pgl1 mutants) is that the time of cell cycle exit would be delayed, which would explain the smaller cells (rather than being a consequence of an altered endocycle). If the authors find no differences in total cell number, this will support their own hypothesis. I would like to emphasize that showing an effect of pectin on the timing of cell cycle exit (rather than on endocycle directly) and thus on cell size, would still be a nice result. On the other hand, publishing the paper in its current form could potentially propagate the false hypothesis that pectin abundance drives the endocycle.

Response: Thank you for your thoughtful suggestion and emphasizing the importance of clarifying the relationship between pectin, ploidy, and cell cycle dynamics. We have addressed your concerns with new experimental data and analyses:

1. To directly test whether pectin influences timing of cell cycle exit in addition to endocycle progression, we performed ploidy analysis and cell size/count measurements across three developmental leaf stages. We sampled the third pair of leaves at 14 (stage 1), 18 (stage 2), and 28 (stage 3) days after sowing, which represent stages with dividing cells, cell cycle exit, and endoreduplicating cells, respectively. In *PGL1-OX* and *pgl1* lines, we observed no significant differences in the epidermal cell number or leaf area at stages 1 and 2 (revised Fig. 5). This indicates that timing of cell cycle exit is not significantly altered in these lines, ruling out a primary effect on mitotic cell cycle duration by *PGL1*. In contrast, at stage 3, when cells are fully committed to endoreduplication, *PGL1-OX* exhibited significantly larger cells (reduced cell

number per unit area) and elevated 16C ploidy while *pgl1* showed smaller cells and reduced 16C ploidy (revised Fig. 5c-f). These stage-specific effects demonstrate that PGL1-mediated pectin degradation specifically promotes endocycle progression rather than altering mitotic cell cycle exit.

2. There are some technical challenges with analyzing *tcpΔ7* leaves. As shown below (Fig. R1 below), its curled leaves and highly dense trichomes (Fig. R1) precluded accurate epidermal cell counting. However, flow cytometry and DAPI staining revealed that *tcpΔ7* retains a high proportion of 2C/4C nuclei even in mature leaves (Fig. 1e; see also Fig. R2 below), suggesting prolonged mitotic activity. While this complicates developmental staging, it further supports our model that persistent mitotic cycling (rather than altered exit timing) in *tcpΔ7* may indirectly suppress endocycle progression, consistent with the role of the downstream PGL1 in promoting pectin degradation to enable cell enlargement.

Fig R1. Morphology of *tcpΔ7* leaves. a-b Cryo-SEM analysis of *tcpΔ7* leaves at stage 1 (a) and stage 2 (b), scale bar, 500 μ m.

Regarding the contribution of endocycle to cell size, it should be noted that while 16C cells are a minority, their disproportionate contribution to organ size is evident. The largest cells (16C) in *PGL1-OX* are ~ 25 x larger than the smallest (2C) (Fig. 4d-e). Critically, the total cell number remains unchanged across genotypes when cells exit the cell cycle (revised Fig. 5d-e). Taken together, these results confirm that differences in final organ size primarily arise from post-mitotic cell enlargement via endocycle regulation.

Overall our data robustly supported the model that pectin degradation by PGL1 promotes endocycle progression and cell enlargement over leaf development. Accompanying the revised Fig. 5, the text was revised accordingly in the revised manuscript (lines 268 to 282).

Additional comments:

1. It is mentioned that the 4C/8C ratio is a surrogate for endocycle onset. This is true when young leaves are used, but not when ploidy measurements are made on 4-week-old plants (as is the case in this paper). As mentioned above, the best way to demonstrate a clear effect on ploidy levels is to measure ploidy levels during the process of leaf development, from a young dividing leaf to a mature leaf.

Response: Thank you for this constructive suggestion. As mentioned in our above responses, this issue was addressed as suggested (revised Fig. 5).

2. Often, data from different tissues or different ages are compared. This is the case for many of the experiments, but to give an example: Figure 1 measures cell size in 7-day-old cotyledons and correlates this with ploidy measurements on 4-week-old rosettes. The most problematic case is the last figure, where ploidy measurements are made on 4-week-old *ccs52a* plants, but pectin measurements are made on 7-day-old seedlings. Using these data to conclude that there is no relationship between ploidy and pectin content is incorrect. Pectin should be measured at the same time and in the same tissue as when ploidy differences are observed.

Response: Thank you for the constructive suggestion on maintaining consistency of the samples in different experiments. As suggested, we have updated measurement of the ploidy levels so that ploidy and pectin were measured in the same stages (see revised Figs. 1, 5 and 9). The experimental conclusions were not impacted by these updates. For example, difference in the endocycle profiles between the wild type and *tcpΔ7* was consistent in the cotyledons and mature leaves (Fig. R2).

Fig R2. CIN-TCPs promote endocycle. **a-b** Representative ploidy profiles obtained by flow cytometry analysis of 4-week-old rosette leaves (a) and 7-day-old cotyledons (b) of wild type and *tcpΔ7*. **c-d** Quantification of EI, the ratio of 8C to 4C nuclei, and the ratio of 16C to 8C nuclei in rosette leaves (c) and cotyledons (d). Values are mean \pm SD ($n = 6$). *, $p < 0.05$; ***, $p < 0.001$ by Student's *t*-test.

3. Line 186: The plates are incorrectly referenced.

Response: Thank you for pointing out this error, which was fixed in the revised manuscript (line 186).